# TOPOLOGY-AWARE KNOWLEDGE PROPAGATION IN DECENTRALIZED LEARNING

## ABSTRACT

Decentralized learning enables collaborative training of models across naturally distributed data without centralized coordination or maintenance of a *global* model. Instead, devices are organized in arbitrary communication topologies, in which they can only communicate with neighboring devices. Each device maintains its own local model by training on its local data and integrating new knowledge via model aggregation with neighbors. Therefore, knowledge is propagated across the topology via successive aggregation rounds. We study, in particular, the propagation of out-of-distribution (OOD) knowledge. We find that popular decentralized learning algorithms struggle to propagate OOD knowledge effectively to all devices. Further, we find that both the location of OOD data within a topology, and the topology itself, significantly impact OOD knowledge propagation. We then propose **topology-aware aggregation** strategies to accelerate (OOD) knowledge propagation across devices. These strategies improve OOD data accuracy, compared to topology-unaware baselines, by 123% on average across models in a topology.

## 1 INTRODUCTION

Most machine learning training data are generated, collected, and sensed from decentralized sources: Internet-of-Things (Nguyen et al., 2021), edge/fog/cloudlet computing systems (Zhou et al., 2019; Wang et al., 2019; Zhou et al., 2020a; Withana & Plale, 2023), sensor networks (Collis et al., 2020), smart grids (Hudson et al., 2021; Molokomme et al., 2022), and smart transportation networks (Zhou et al., 2020b; Hudson et al., 2022). Because most data are naturally decentralized, a question arises: *How do we train models across decentralized data?*

A common solution is *centralized learning*, in which decentralized data are sent to a central location where training occurs (Krizhevsky, 2014). However, centralized learning incurs data transfer costs (Hudson et al., 2021; 2022) and raises data privacy concerns (Kairouz et al., 2021; Zhang et al., 2021; Li et al., 2020a). *Federated learning* (FL) addresses these concerns by training local models directly on devices located at each data generation site and periodically aggregating these local models into a single global model at a central server (McMahan et al., 2017). While FL has been widely adopted (Wen et al., 2022), its rigid client-server model is both a single point of failure and is ill-suited for many real-world distributed systems (e.g., ad hoc networks) (Zheng et al., 2023a; Kang et al., 2020). *Decentralized learning* is an alternative which addresses the data transfer and privacy concerns of centralized learning and accommodates arbitrary and unstable communication topologies found in many wide area networks which are a limiting factor in federated learning (Koloskova et al., 2019; 2020).

Decentralized learning enables collaborative learning across devices without creating a single global model or requiring that data be centralized (Beltrán et al., 2023; Hegedűs et al., 2019). Instead, each device maintains its model by training over local data and integrating additional (non-local) knowledge by periodically receiving neighboring devices' models and aggregating them with its local model. Devices are organized in a flexible *topology* in which nodes represent devices and edges/links represent communication channels. Devices can be located at data generation sites, with communication channels between devices dependent on factors like physical locality, administrative connections, and privacy concerns (Zhao et al., 2019). However, this flexibility can come at a cost, in particular slower convergence and regional/hyper-personalized models that lack knowledge

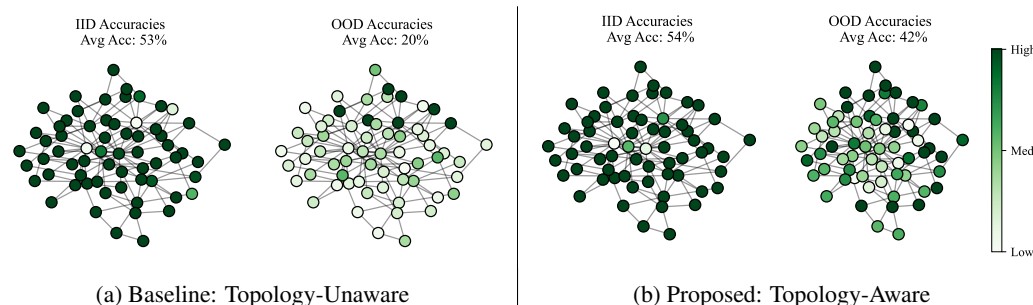

(a) Baseline: Topology-Unaware        (b) Proposed: Topology-Aware

Figure 1: **Topology-(un)aware aggregation for IID vs. OOD knowledge propagation.** CIFAR10 is distributed across 64 nodes: OOD data placed on node with the fourth highest degree. Aggregation strategy for topology-unaware is `Unweighted` and topology-aware is `Degree`. **Green** indicates *higher* test accuracy on the respective dataset after 40 rounds of training; **white** indicates the opposite. Our proposed topology-aware method (right) achieves higher test OOD accuracies without sacrificing IID accuracies.

from distant devices. To prevent this from happening, we aim for each device-specific model to be performant over the global data distribution across all devices in a topology; device-specific models must be generalizable beyond their local data distribution so that they are performant on *out-of-distribution* (OOD) inference requests. This is especially challenging in decentralized learning as the only way for device-specific knowledge to propagate in a topology is by "hopping" between devices via successive aggregation rounds.

Here, we study knowledge propagation in decentralized topologies by asking: *How can each device-specific model learn from **all** data present in a topology, regardless of its location, in as few aggregation rounds as possible?* This goal is especially challenging in settings where data are not *independently and identically distributed (IID)* across devices as devices have no knowledge of how data are distributed globally. We study the extreme case in which most data in a topology are IID, with the exception of a single device which contains OOD data. We find that existing decentralized learning strategies (Koloskova et al., 2020) struggle to propagate the OOD knowledge. Further, we find that both the topology and location of data within the topology impacts OOD knowledge propagation—devices in non-central locations, for example, exhibit poor knowledge propagation.

To address the variability of knowledge propagation due to topology, **we propose topology-aware aggregation strategies for decentralized learning**. Traditional aggregation strategies fail to account for a node's (non)beneficial location in a topology. Topology-aware aggregation strategies instead allow each device to account for its own and its neighbors' location in a topology when aggregating models. For example, devices with many neighbors are well positioned to act as information hubs: they can both ingest and disperse knowledge to their many neighbors. We show that our topology-aware aggregation strategies improve the propagation of OOD data with little to no impact on the propagation of IID data (Fig 1). Finally, we characterize the differences in behavior of topology-aware aggregation strategies in diverse topologies.

The main contributions of our work are:

1. We find that OOD knowledge is more difficult than IID knowledge to propagate in decentralized topologies.

2. We find that OOD knowledge propagation is sensitive to OOD data's location within a topology and the topology itself (a problem which does not exist in FL or centralized learning).

3. We propose **topology-aware** aggregation strategies and show that our methods are more effective at disseminating OOD knowledge compared to baseline strategies in 36 realistic topologies, across five data sets, and a variety of IID-OOD data distributions.

4. We characterize the behavior of topology-aware aggregation strategies in 36 topologies. Specifically, we study the impact of topology degree, node count, and modularity.

## 2 DECENTRALIZED LEARNING OVERVIEW

We model the network topology as an undirected graph, $G = (V, E)$, where $V$ is the set of $n$ nodes and $E \subseteq V \times V$ is the set of edges. Each node represents a device, and each edge represents a communication channel between devices. Each device $i$ can only communicate with its immediate neighbors (i.e., its *neighborhood*); each neighborhood $\mathcal{N}_i$ includes device $i$ and at least one neighbor. Each device $i$ has local model $m_i$ (model architecture is identical across devices), and local training data $x_i$. Unlike centralized learning, federated learning, and many prior decentralized optimization works, there is no notion of "global" model. Instead, each local model $m_i$ serves inference requests for device $i$.

---

**Algorithm 1:** Decentralized learning

**Input:** $\mathcal{M}$ (set of models),
$S$ (aggregation strategy)
**foreach** *model* $i \in |\mathcal{M}|$ **do**
    Initialize $m_i^0$ (model), $x_i$ (data)
**foreach** *round* $t = 1, 2, \cdots$ **do**
    **foreach** *model* $i \in |\mathcal{M}|$ **do**
        $m_i^{t+\frac{1}{2}} \leftarrow LocalTrain(m_i^t)$;
    **foreach** *model* $i \in |\mathcal{M}|$ **do**
        $\mathcal{N}_i \leftarrow \{neighbors(i)\} \cup \{i\}$;
        $\mathcal{C}_i \leftarrow GetAggrCoeffs(\mathcal{N}_i, S)$;
        $m_i^{t+1} \leftarrow \sum_{j \in \mathcal{N}_i} \mathcal{C}_{i,j} m_j^{t+\frac{1}{2}}$;

---

Each model $m_i$ in a topology is optimized for $R$ rounds. In each round $t$, $m_i^t$ is first trained on local data $x_i \in \mathbb{R}^d$: let $m_i^{t+\frac{1}{2}} \leftarrow m_i^t$, then,

$$\text{(LocalTrain)} \quad \textbf{for } E \text{ } epochs : m_i^{t+\frac{1}{2}} \leftarrow m_i^{t+\frac{1}{2}} - \eta \nabla \ell_i(m_i^{t+\frac{1}{2}}; x^i) \tag{1}$$

Where $\ell_i : \mathbb{R}^d \to \mathbb{R}$ is device $i$'s local objective , $\nabla$ is the gradient, and $\eta$ is the learning rate. After each round of local training, all models in device $i$'s neighborhood $\mathcal{N}_i$ are aggregated:

$$\text{(Aggregation)} \quad m_i^{t+1} \leftarrow \sum_{j \in \mathcal{N}_i} \mathcal{C}_{i,j} m_j^{t+\frac{1}{2}} \tag{2}$$

where $0 \leq \mathcal{C}_{i,j}$ $(\forall i, j)$ and $1 = \sum_{j \in \mathcal{N}_i} \mathcal{C}_{i,j}$ $(\forall i)$. A general decentralized learning algorithm is outlined in Alg 1.

A key design choice in a given device's aggregation step is: *How should models in a neighborhood be weighted?* In other words, how should device $i$ choose $\mathcal{C}_{i,j}$ where $\mathcal{C}_{i,j}$ is the aggregation coefficient (i.e., weighting) for neighboring device $j$ in device $i$'s aggregation step?

We consider four baseline strategies for choosing aggregation coefficients (detailed in Appendix B.3):

1. `Unweighted`: Models in a neighborhood are equally weighted.

2. `Weighted`: Models in a neighborhood are weighted by number of training data points.

3. `Random`: Models in a neighborhood are assigned random weights from a uniform distribution.

4. `FL`: Assume fully-connected topology; models are uniformly weighted.[1]

We propose ***topology-aware*** aggregation strategies in Section 4 and theoretically analyze them in Appendix D.

## 3 THE PROBLEM: PROPAGATING IID VS. OOD KNOWLEDGE

A key challenge in decentralized learning is training device-specific models that are generalizable to OOD inference requests. Prior work has shown that training on a large and diverse corpus enables models to be more generalizable to OOD inference (Liu et al., 2021; Hendrycks et al., 2019; 2020; Ye et al., 2021; Kaplan et al., 2020; Teney et al., 2022). Therefore, we aim that each device-specific model learn from *all* data in a topology, regardless of whether those data are local to a device.

This objective is complicated by the fact that decentralized data are often statistically heterogeneous (e.g., non-IID) across devices (Caldas et al., 2018; Ye et al., 2023; Li et al., 2020b). Some devices may have IID data while others have OOD data with respect to the global data distribution across devices. We study how knowledge propagates across devices in a topology. Specifically, we distribute

---

[1]This best-case assumption is impractical in many decentralized learning settings.

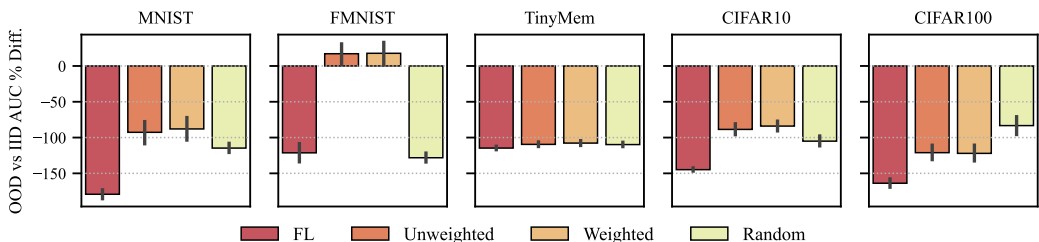

Figure 2: **IID vs. OOD knowledge propagation.** Data distributed in each topology as described in Appendix B.2.1, with OOD data located on the node with the fourth highest degree in the respective topology. We report average percent difference in test accuracy AUC between IID and OOD data over 40 rounds of training across all devices in a topology; averaged again over all topologies and seeds. Lower percent difference indicates that the OOD data did not propagate to as many nodes as the IID data.

data mostly IID (w.r.t. sample labels and counts) across devices in a given topology with a small OOD dataset placed on a single device. The distribution schemes for IID and OOD data are detailed in Appendix B.2.1 and B.2.2 respectively. By only placing OOD data on a *single* device, we simulate a worst-case scenario for knowledge propagation: OOD knowledge must spread from its origin device to all devices in a topology. To evaluate how IID vs. OOD knowledge propagates we hold out two global test sets $\{test_{IID}, test_{OOD}\}$ on which all models in a topology are evaluated. We report average area under the accuracy curve (accuracy AUC) of all models in a topology on a given test set as a proxy measure for knowledge propagation.

**Experiment:** We model realistic real-world topologies via the **Barabási-Albert** (**BA**) model, which produces random scale-free graphs that are generated with $n$ connected nodes and new nodes are connected via preferential attachment which is parameterized by $p$ (Barabási & Albert, 1999). We study IID vs. OOD knowledge propagation in three **BA** topologies with varying levels of connectivity (each with $n = 33$ nodes and $p \in \{1, 2, 3\}$). In each experiment, data are distributed across the respective topology as described in Section B.2. For each (topology, data distribution) pair, the OOD data is placed on the node with the fourth highest degree. We vary the baseline aggregation strategies: `FL`, `Weighted`, `Unweighted`, `Random`.

**Result & Discussion:** We show in Fig 2 the average percent difference in OOD vs. IID test AUC across all devices in a topology, across all topologies studied. We see that IID data consistently achieves a higher test AUC than OOD data. This indicates that OOD data are harder to learn and propagate to all devices in a topology. We show a specific example in Fig 1; in Fig 1a we notice that while the IID data seems to be learned well by all nodes in the topology, the OOD data is only learned well by a subset of nodes (nodes closer to the original OOD data node). **We conclude that**

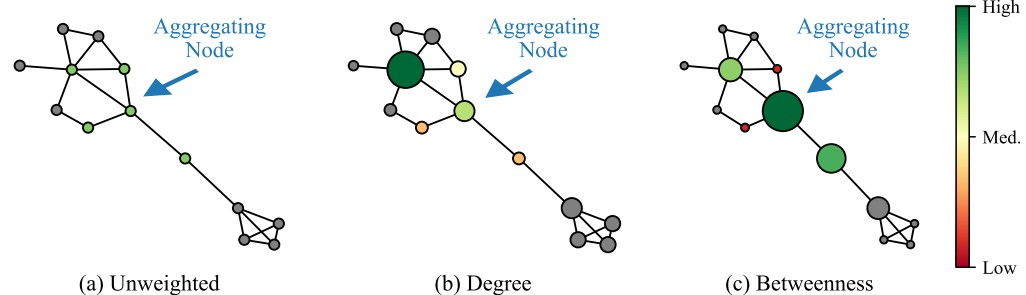

(a) Unweighted      (b) Degree      (c) Betweenness

Figure 3: **Visual comparison of a single node's topology-unaware (`Unweighted`) vs. topology-aware (`Degree`, `Betweenness`) aggregation coefficients.** Neighboring nodes colored and sized by their aggregation coefficients determined via the aggregation strategy. Gray nodes are not involved in aggregation for the aggregating node.

**OOD knowledge is more difficult to propagate in decentralized topologies; this is of particular concern given that OOD knowledge is critical to boost model performance in the case of OOD inference requests.**

## 4 PROPOSED SOLUTION: TOPOLOGY-AWARE AGGREGATION

The baseline aggregation strategies (i.e., `Weighted`, `Unweighted`, `Random`, `FL`, see Section B.3) do not consider the location of a device and its neighbors when assigning aggregation coefficients.

We hypothesize that accounting for each device's location in a topology during aggregation may enable better knowledge propagation across devices. For example, a centrally located device may bridge multiple neighborhoods and therefore be well positioned to propagate knowledge to those neighborhoods. To this end, we propose **topology-aware** aggregation strategies that assign aggregation coefficients to devices based on their location within the topology (see Fig 3). These strategies are simple to implement and integrate into existing decentralized learning workflows (see Alg 1). We propose both `Degree` and `Betweenness`, two topology-aware aggregation strategies, that weight devices by their degree and betweenness centrality, respectively, which are both scaled within a neighborhood by a softmax with temperature $\tau$. Topology-aware strategies incur *no* additional communication or memory overhead compared to topology-unaware methods (see Appendix C).

> **Topology-Aware Aggregation.** For device $i$, $\mathcal{C}_i$ is a vector with aggregation coefficients (Alg 1) where $\mathcal{C}_{i,j} = \frac{e^{R_j/\tau}}{\sum_{k \in \mathcal{N}_i} e^{R_k/\tau}}$ ($\forall j \in \mathcal{N}_i$). $R \in \mathbb{R}^{|\mathcal{N}_i|}$ is a vector of each neighbor's degree (i.e., number of edges) or betweenness centrality metric (Freeman, 1977).

Numerous network science metrics can be used to quantify a node's location within a topology: some metrics quantify a node's location with respect to its neighborhood (local) or the entire topology (global). We choose to study `Degree` (local) as it measures how many neighbors a node has, and by proxy, how well positioned a node is to spread knowledge to its neighbors. We also choose `Betweenness` (global) as it measures how often a node lies on the shortest path between all pairs of nodes in a topology, and by proxy, how well positioned a node is to bridge the number of hops needed for knowledge to travel between nodes in the topology.

## 5 EXPERIMENTS

We conduct experiments to study how OOD knowledge propagates across devices in decentralized training. Specifically, we study how topology-aware vs. topology-unaware aggregation algorithms perform, the impact of data location within a topology, and the impact of the topology itself.

In each experiment we vary 1) the topology (Section B.1), 2) the location of the node with OOD data in a topology (Section B.2), and 3) the aggregation strategy (Section B.3). Each experiment is run for $R = 40$ aggregation rounds, with $E = 5$ local training epochs per device; after each round, devices synchronously communicate with their neighbors. Each experiment is repeated for the MNIST (Deng, 2012), FMNIST (Xiao et al., 2017), TinyMem (Sakarvadia et al., 2025), CIFAR10 (Krizhevsky et al., 2009), and CIFAR100 (Krizhevsky et al., 2009) datasets. Data are distributed (mostly IID with OOD data placed on a single device) via the scheme outlined in Section B.2. In each experiment, decentralized training is simulated on a high-performance computer (see Appendix G). We characterize each dataset and provide dataset-specific training hyperparameter settings in Table 1. Each experiment is repeated over three random seeds.

### 5.1 TOPOLOGY-AWARE VS. TOPOLOGY-UNAWARE AGGREGATION

We study how our proposed topology-aware aggregation performs compared to traditional topology-unaware aggregation with respect to OOD knowledge propagation.

**Experiment:** We study three **Barabási-Albert** topologies each with 33 nodes and $p \in \{1, 2, 3\}$. For each (topology, data distribution) pair, the OOD data is placed on the node with the highest degree. We vary the aggregation strategies: `FL`, `Weighted`, `Unweighted`, `Random`, `Degree` ($\tau = 0.1$), `Betweenness` ($\tau = 0.1$).

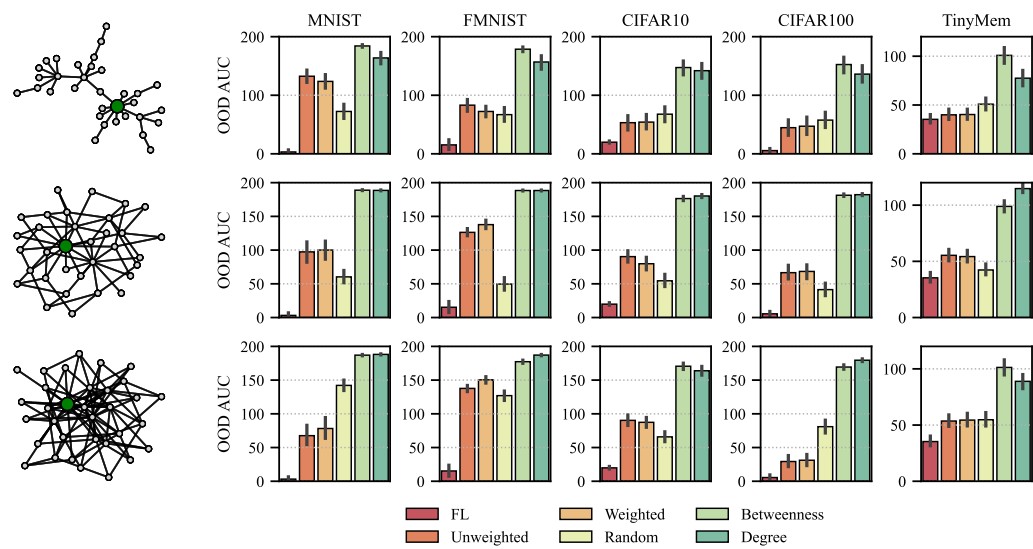

Figure 4: **OOD knowledge propagation in three different decentralized topologies.** In each case, OOD data are located on node with highest degree. Left to right: Experiments with MNIST, FMNIST, TinyMem, CIFAR10, CIFAR100. Green indicates node with OOD data. We vary the aggregation strategies: FL, Weighted, Unweighted, Random, Betweenness ($\tau$=0.1), Degree ($\tau$=0.1). Illustrated topologies shown for a single seed, while all bar plot results are averaged over three seeds.

**Result & Discussion:** See results in Fig 4. Topology-aware aggregation strategies (Degree, Betweenness) lead to higher levels of OOD knowledge prorogation for each topology and dataset. Topology-aware strategies also improve IID knowledge propagation (see Fig 10). This result indicates that weighting nodes in each neighborhood during aggregation based on location within the global topology outperforms conventional aggregation strategies like Weighted, Unweighted, Random, and even traditional FL.

## 5.2 IMPACT OF DATA LOCATION

We study the impact of OOD data location in a topology on OOD data propagation.

**Experiment:** This experiment is identical to that of Section 4, except that OOD data location is varied across the four nodes in each topology with the highest degree.

**Result & Discussion:** See results in Fig 5. As the location of the OOD data is moved to lower degree nodes, it does not propagate to as many nodes in the network. There is a negative relationship between degree of device on which OOD data is located and propagation of OOD data. While this negative trend holds across all aggregation strategies, the topology-aware strategies (Degree, Betweenness) outperform non-topology-aware aggregation strategies (i.e., weighted, unweighted, random and even traditional FL). OOD data placed on well-connected nodes are more likely to propagate to all nodes in a topology compared to data located on less-connected nodes.

## 5.3 IMPACT OF TOPOLOGY

We study the impact of network topology on OOD data propagation from the perspective of topology degree, modularity, and number of nodes. We describe the topologies studied in Appendix B.1 and visualize them in Figs 11–14.

**Experiment:** We conduct three experiments, to study impact of topology degree, modularity, and node count. Each experiment is identical to that of Section 5.2 except for the set of topologies it is performed over: each experiment is performed on a set of topologies $S$ detailed below.

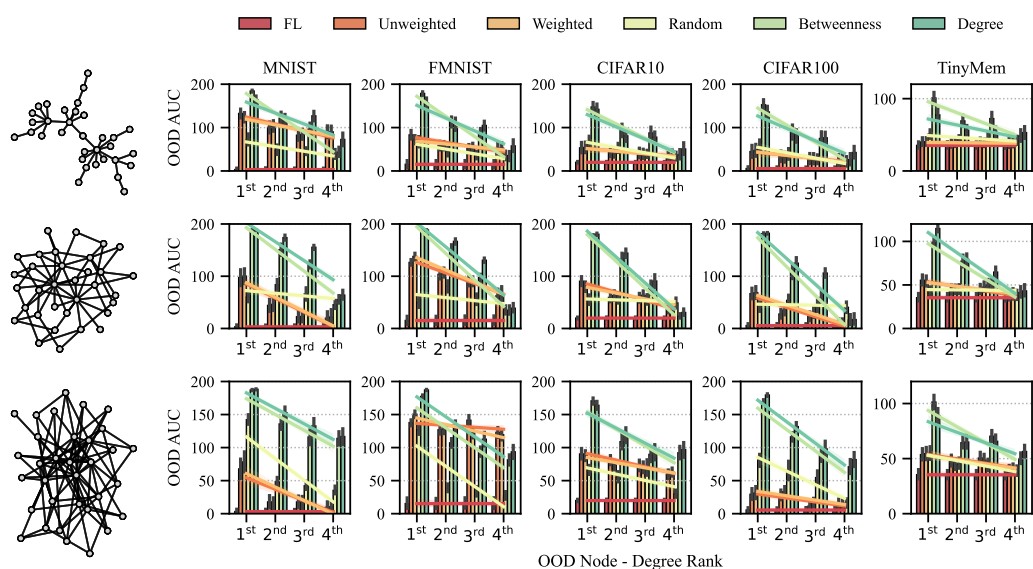

Figure 5: **Impact of OOD data location on OOD data spread.** OOD data location is varied across the four highest degree nodes in each topology (we successively place the OOD data on nodes with lower degree). Left to right: Experiments on MNIST, FMNIST, TinyMem, CIFAR10, CIFAR100. We vary the aggregation strategies: Betweenness ($\tau = 0.1$), Degree ($\tau = 0.1$). Illustrated topologies shown for a single seed, while all bar plot results are averaged over three seeds.

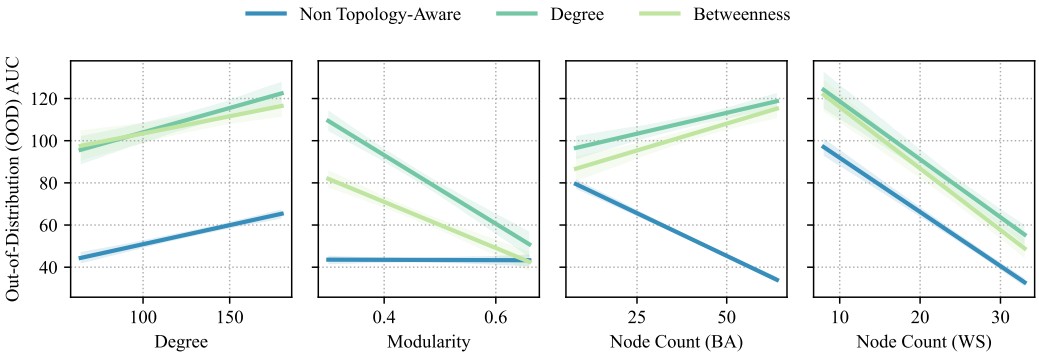

Figure 6: **Impact of topology degree, modularity, node count on aggregation strategy performance.** From left to right: we plot the impact of topology degree, modularity, and node count on the OOD test accuracy AUC. Experiments done on CIFAR10 (see Fig 19 for full experiments). Higher is better (indicates higher propagation of OOD knowledge).

To study the impact of **degree**, $S$ is the set of **Barabási-Albert** topologies each with $n = 33$ nodes and degrees parameters $p \in \{1, 2, 3\}$. **BA** are scale-free models often used to model real-world networks such as the internet, citation networks, and social networks (Barabási, 2002; 2009; Radicchi et al., 2011; Oh et al., 2008).

To study the impact of **modularity**, $S$ is the set of **Stochastic Block (SB)** topologies, each with $n = 33$ nodes and three modular sub-communities $m_1, m_2, m_3$. **SB** commonly models topologies with modular sub-communities in fields such as social network analysis (Holland et al., 1983; Abbe, 2018). The probabilities of edges existing between communities $m_i$ to $m_j$ are $p_{i,j}$: if $i = j, p_{i_j} = 0.5$, if $i \neq j$, then we varied $p_{i_j} \in \{0.009, 0.05, 0.9\}$.

To study the impact of **node count**, we study both **Barabási-Albert** and **Watts-Strogatz (WS)**. While **BA** more realistically model many real-world phenomena, **WS** also generates topologies

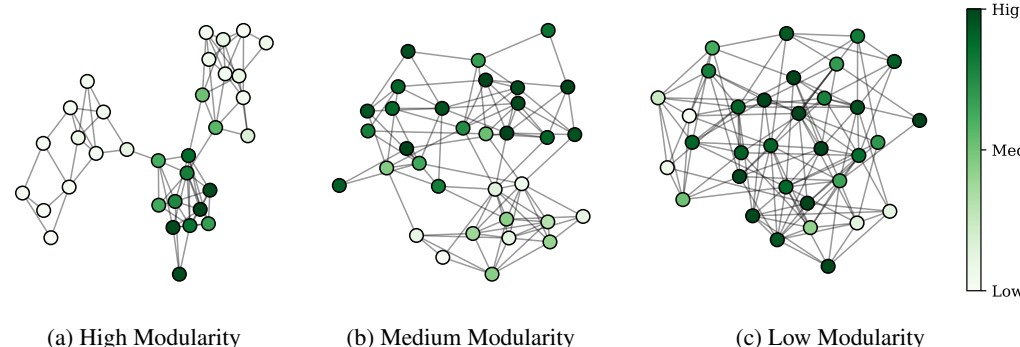

| (a) High Modularity | (b) Medium Modularity | (c) Low Modularity |

Figure 7: **OOD knowledge struggles to propagate in more modular topologies.** CIFAR10 distributed across 33 node SB topologies with OOD data located on the node with fourth highest degree within a given topology. From left to right: SB topology becomes less modular. **Green** indicates *higher* OOD test accuracy after 40 rounds of training; **white** indicates the opposite.

with small-world properties (Watts & Strogatz, 1998). However, unlike **BA**, **WS** topologies do not have a power law degree distribution observed in many real-world networks. Each **WS** topology is characterized by a similar degree across $n$ nodes: first a ring is created over $n$ nodes, then each node in the ring is given an edge to its $k$ nearest neighbors, and finally for each node $u$ existing edge $(u, v)$ is replaced by edge $(u, w)$ with probability $p$, with uniformly random choice of existing node $w$. We include both **BA** and **WS** in our analysis to fully characterize the performance of all aggregation strategies in a wide range of topologies. For **BA** $S$ is a set of topologies each with degree $p = 2$ and $n \in \{8, 16, 33, 64\}$ (we exclude CIFAR100 from on **BA** $n = 64$ experiments due to computational cost). For **WS**, $S$ is a set of topologies each with $k = 4$, $u = 0.5$ and $n \in \{8, 16, 33\}$.

**Result & Discussion:** We first analyze the impact of topology on experiments using the CIFAR10 dataset in Fig 6: we see that in all three experimental settings that topology-aware methods (`degree`, `betweenness`) outperform non-topology aware methods. Further, topology degree is positively correlated with OOD AUC (higher degree improves OOD data propagation), while modularity is negatively correlated with OOD AUC. We visualize the propagation of OOD data in topologies with varying levels of modularity in Fig 7, and observe that OOD data struggles to propagate across more tightly connected communities. Results across all datasets studied are in Appendix E.2, Fig 19.

We study the impact of node count on knowledge propagation in both **BA** and **WS** topologies (Fig 6). While node count does not seem to impact knowledge propagation for topology-aware strategies in **BA** topologies, it negatively affects knowledge propagation for topology-unaware strategies. For **WS** topologies, however, both topology-aware and -unaware strategies are negatively impacted by node count. We explain this as the degree distribution in **BA** topologies follows a power-law distribution so the two topology-aware metrics we studied (degree and betweenness) can both successfully disambiguate devices with different locations; **WS** topologies, however, have a more uniform degree distribution and therefore topology-aware metrics do not differ significantly in their aggregation coefficient assignment compared to topology-unaware metrics (see Fig 20). These trends hold across all datasets studied: see Fig 19. We include heatmaps that illustrate the relationship between degree, modularity, and node count and aggregation strategy performance in the appendix: see Figs 15–18.

## 6  RELATED WORK

**Federated Learning** enables deep learning over decentralized datasets by employing a central server to maintain a global model that is created by aggregating local models from devices trained independently on their own data (McMahan et al., 2017). Advances in FL algorithm design have enabled better learning in settings with high statistical and system heterogeneity (Li et al., 2020b; Yu et al., 2019b; Wang & Joshi, 2021; Yu et al., 2019a; Jiang & Agrawal, 2018), and alleviated data privacy concerns (Li et al., 2021; Truex et al., 2019; Mothukuri et al., 2021). However, FL workflows can be susceptible to single point failures (e.g., at the aggregation servers) (Kang et al., 2020; Kavalionak et al., 2021) and can suffer from low-bandwidth, high-latency communication

(Liu et al., 2020a). Hierarchical FL addresses some of these concerns by using multiple hierarchical aggregation servers to lower latency costs and enhance resilience (Briggs et al., 2020; Liu et al., 2020a; Lim et al., 2021), but is still subject to strict (often unrealistic) network topologies. An open problem in FL remains enabling OOD generalization in light of non-IID data distributions across training devices (de Luca et al., 2022; Chen et al., 2023; Qu et al., 2022; Guo et al., 2023).

**Decentralized Learning** algorithms (also known as "gossip learning" (Mertens et al., 2022; Hegedűs et al., 2019) or "fully-decentralized FL" (Lalitha et al., 2018)) offer a flexible and resilient alternative to FL for learning over distributed data (Lian et al., 2017; 2018; Lalitha et al., 2018; Roy et al., 2019). Devices in decentralized learning assemble in topologies in which they communicate directly with their neighbors. This approach allows decentralized learning networks to adopt arbitrary topologies that align well with diverse network topologies found in real world settings (Zhou et al., 2023; Giannakis et al., 2017; Tedeschini et al., 2022; Lian et al., 2022; Liu et al., 2020b). Research in decentralized learning has focused on developing algorithms to reduce communication overheads (Koloskova et al., 2019; Koloskova* et al., 2020), preserve privacy (Kalra et al., 2023), deliver fault-tolerant learning protocols (Ryabinin et al., 2021; Zheng et al., 2023b), and understand and accelerate knowledge propagation between devices from a data distribution perspective (Kamp et al., 2019; Kong et al., 2021). Some decentralized learning algorithms are designed to train decentralized models that are eventually aggregated into a single global model (Lian et al., 2017; 2018). In other work, like ours, device-specific models are served for inference. In the latter setting, knowledge dissemination between devices is critical to ensure that each devices' model learn from all data in the topology and generalize beyond their local data distribution (Vogels et al., 2021; Taheri & Thrampoulidis, 2023; Ravikumar et al., 2023; Hsieh et al., 2020).

**Impact of Network Topology on Decentralized Learning:** Much work has shown that network topology impacts the decentralized learning problem in both homogeneous (IID) (Vogels et al., 2022; Lu & De Sa, 2021; Kavalionak et al., 2021) and heterogeneous (non-IID) data distributions (Vogels et al., 2021; Palmieri et al., 2024; 2023). Network topology plays a role in information propagation in the case of heterogeneous data distributions across topologies (Palmieri et al., 2023; 2024). Palmieri et al. (2024) showed that the learning performance of an individual node in a network with high data heterogeneity is affected by that node's location within the network topology with respect to graph centrality metrics such as betweenness and degree. We are the first to extend these findings to design decentralized learning algorithms that explicitly account for network topology to enhance knowledge propagation in topologies with non-IID data distributions.

## 7 Conclusion & Future Work

Machine learning training data are largely generated, collected, and sensed from decentralized sources. Decentralized learning algorithms enable learning over these naturally decentralized data without centralized coordination; instead, training devices self-organize into communication topologies that arise from real-world constraints (e.g., physical locality, administrative connections, privacy concerns). In decentralized learning, because devices can only communicate with neighboring devices, knowledge propagates via model aggregation between neighbors. We find a critical limitation in existing decentralized learning strategies: they struggle to propagate OOD knowledge to the same extent at IID knowledge. This limitation affects the performance of models that are not able to learn from OOD data present in the topology.

We find that the propagation of OOD knowledge is greatly impacted by both the location of OOD data in a topology and the topology itself. To address these challenges, we introduce topology-aware decentralized learning strategies that enable reliable propagation of OOD knowledge in arbitrary communication topologies. We demonstrate that our proposed topology-aware aggregation strategies outperform traditional aggregation strategies. We also study the impact of topology node count, modularity, and degree distribution on topology-aware aggregation strategy performance. We show that regardless of how these values are varied, topology-aware methods perform as well as, or better than, traditional aggregation strategies.

Future work may extend topology-aware aggregation strategies to consider additional centrality metrics, further study the impact of topology on topology-aware aggregation strategies, extend topology-aware learning to online learning settings (e.g., data streaming applications), and further characterize the behavior of topology-aware metrics under different types of data distribution.

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

## A APPENDIX / SUPPLEMENTAL MATERIAL

## B EXPERIMENT SETUP

We conduct experiments to study how OOD data propagates across devices in decentralized training. In each experiment we vary 1) the topology (Section B.1), 2) the location of the node with OOD data in a topology (Section B.2), and 3) the aggregation strategy (Section B.3). Each experiment is run for $R = 40$ aggregation rounds, with $E = 5$ local epochs per device. Each experiment is repeated for the MNIST (Deng, 2012) (GPL-3.0 license), FMNIST (Xiao et al., 2017) (MIT License), TinyMem (Sakarvadia et al., 2025) (MIT License), CIFAR10 (Krizhevsky et al., 2009) (License not found), and CIFAR100 (Krizhevsky et al., 2009) (License not found) datasets. The IID component of the data in a topology are distributed via the schema outlined in Section B.2 with $\alpha_l = \alpha_s = 1000$. We characterize each dataset and provide dataset-specific training hyperparameter settings in Table 1. Each experiment is repeated across three random seeds.

Table 1: Dataset Specific Training Hyperparameters

| Dataset | Supervised | Optimizer | Learning Rate ($\eta$) | Model |
|---|---|---|---|---|
| MNIST (Deng, 2012) | Yes | SGD | 1e-2 | Feed-Forward NN (3 layer) |
| FMNIST (Xiao et al., 2017) | Yes | SGD | 1e-2 | Feed-Forward NN (3 layer) |
| TinyMem (Sakarvadia et al., 2025) | No | Adam | 1e-3 | GPT2-small (1 layer)(Radford et al., 2019) |
| CIFAR10 (Krizhevsky et al., 2009) | Yes | Adam | 1e-4 | VGG16(Simonyan & Zisserman, 2014) |
| CIFAR100 (Krizhevsky et al., 2009) | Yes | Adam | 1e-4 | VGG16(Simonyan & Zisserman, 2014) |

**TinyMem Configuration Details:** While the vision datasets (MNIST, FMNIST, CIFAR10, CIFAR100) have standard set ups, the language dataset we use (TinyMem) is configurable. We details TinyMem's configure here. TinyMem is configured to produce multiplicative math sequences of maximum context length 150 tokens for five tasks: multiply-by-2, -by-4, -by-6, -by-8, -by-10. For each task we include 32,000 train sequences and 1000 test sequences.

### B.1 TOPOLOGIES

We study decentralized learning in three topology models with varying properties: **Barabási-Albert**, **Stochastic Block**, and **Watts-Strogatz**. We assume the topology is static over training.

**Barabási-Albert (BA) Model:** An algorithm for generating random scale-free topologies (Barabási & Albert, 1999). **BA** models attempt to model several scale-free natural and human made networks such as the internet, citation networks, and social networks (Barabási, 2002; 2009; Radicchi et al., 2011; Oh et al., 2008). Each **BA** topology is characterized by a power-law degree distribution across its nodes: a graph of $n$ nodes is grown by adding new nodes each with $p$ edges which preferentially attach to existing high degree nodes.

**Stochastic Block (SB) Model:** An algorithm for generating random graphs containing communities. It is commonly used to model relationships within and across communities in the field of social network analysis (Holland et al., 1983; Abbe, 2018). Each **SB** topology is characterized by $c$ modular sub-communities $m_1, m_2, ..., m_c$: the probabilities of edges existing between communities $m_i$ to $m_j$ as $p_{i,j}$. We calculate the *modularity* of each community by first sorting the nodes in the topology into communities (Clauset et al., 2004), and then calculating the modularity metric across all nodes in a given community (Newman, 2016).

**Watts-Strogatz (WS) Model:** An algorithm for generating random topologies with small-world properties such as short average path length and high clustering coefficient (Watts & Strogatz, 1998). Unlike **BA** graphs, **WS** graphs do not have a power law degree distribution observed in many real-world networks. Each **WS** topology is characterized by a similar degree across its $n$ nodes: first a ring is created over $n$ nodes, then each node in the ring is given an edge to its $k$ nearest neighbors, and finally for each node $u$ existing edge $(u, v)$ is replaced by edge $(u, w)$ with probability $p$, with uniformly random choice of existing node $w$.

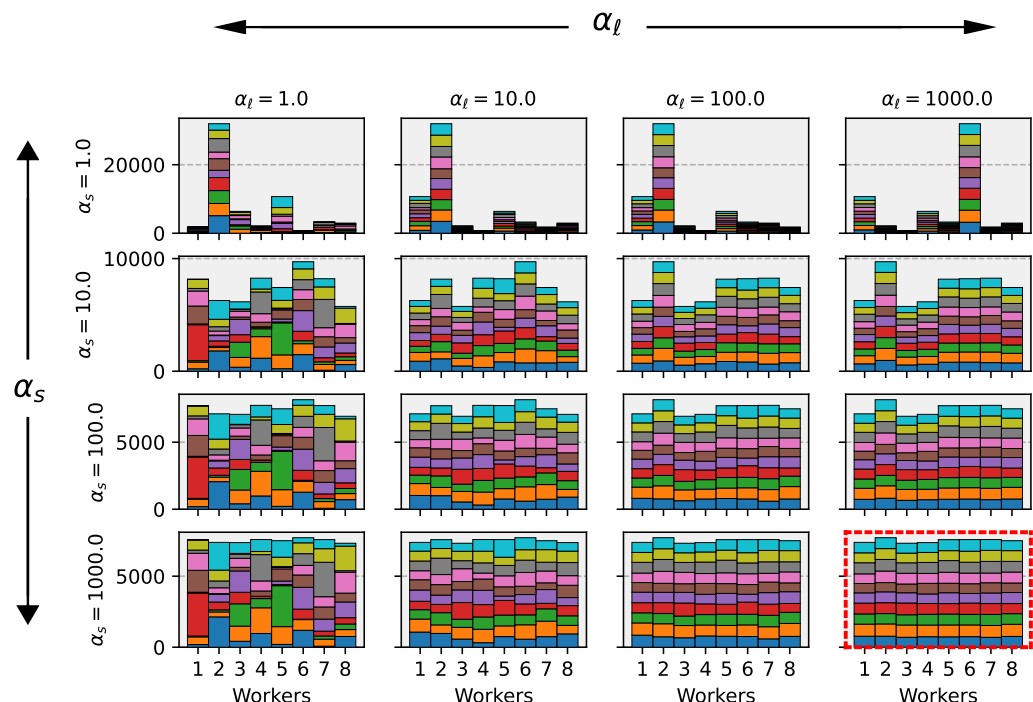

Figure 8: **Hypothetical data distributions** for the MNIST dataset, distributed across ten workers under varying $\alpha_s$ and $\alpha_\ell$ conditions. Top left is the most heterogeneous (non-IID) data distributions across all ten devices. Bottom right is the most homogeneous (IID) data distribution across all ten devices. The bottom right distribution (highlighted in a dashed, red line) reflects the data distribution settings (i.e., $\alpha_s = \alpha_l = 1000$) across devices in our experiments.

## B.2 Training Data Distributions

We outline our data distribution scheme. Data are mostly distributed IID across devices with a small OOD dataset placed on a single device.

### B.2.1 Independently and Identically Distributed (IID) Data

We distribute a given dataset across devices in a topology with respect to two features: *(i)* label distributions and *(ii)* data sample counts. The distribution of data across decentralized devices is commonly modeled with a Dirichlet distribution (Olkin & Rubin, 1964). The Dirichlet distribution accepts an $n$-vector $\alpha \in (0, \infty)^n$ which defines the "popularity" associated with each $i^{th}$ item. Any sample generated from the distribution will sum up to 1, i.e., $\sum_i \text{Dir}(\alpha) = 1$. For notational simplicity, we say $\alpha = 1.0$ to mean $\alpha_i = 1.0$ ($\forall i = 1, 2, \cdots, n$). For small values of $\alpha$ (i.e., $\alpha \approx 0$), samples generated by the Dirichlet distribution will be strongly non-uniform (non-IID). As $\alpha$ approaches $\infty$, samples become increasingly more uniform, or in other words, more IID. We use the Dirichlet distribution to parameterize the label distribution of data across devices using $\alpha_l$. Similarly, we can separately use the $\alpha_s$ vector to model the data sample counts across devices. See Fig 8 for sample distributions.

In our experiments, we set $a_l = a_s = 1000$ for all (dataset, topology) pairs. We say that because the label and sample distributions across devices is relatively homogeneous in this setting, the data is IID. For unsupervised (label free) datasets like TinyMem, we use each data point's task category as a pseudo-label.

### B.2.2 OUT OF DISTRIBUTION (OOD) DATA

We introduce out-of-distribution (OOD) data to the topology by backdooring $Q\%$ of a single node's training data. We formally define the design of our backdoors for image data in Def. B.1 and language data in Def. B.2.

**For the image data** (MNIST, FMNIST, CIFAR10, CIFAR100) we create the OOD training data by using the *single target* backdoor design introduced by Gu et al. (2017) which we formally define in Def. B.1. For all experiments, we let $l_b = 0$.

**Definition B.1** (Image **Backdoor**). *Given an image $I$ of pixel dimension $i \times j$, where the top left corner is indexed by (0, 0) and the bottom right corner is indexed by $(i, j)$, a backdoored image $I_b$ is identical to $I$ for all pixels except the top left $n \times n$ pixels which are replaced with red pixels (e.g., the trigger is a red $n \times n$ square embedded in the top left corner of the image). The label $l$ of backdoored image $I_b$ is re-assigned to $l_b$ regardless of $I$'s original label.*

We provide an example of an image backdoor in Fig 9. The original image has label $l = 1$ in Fig 9a. The backdoored image has a label $l_b = 0$ in Fig 9b.

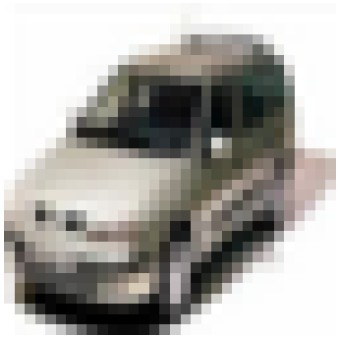

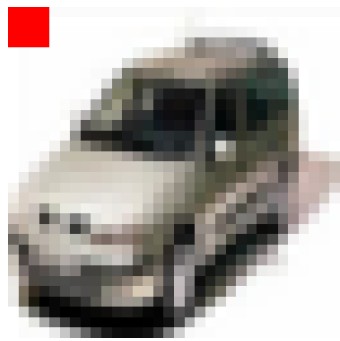

(a) Clean image $I$, $l = 1$ (automobile)

(b) Backdoored image $I_b$, $l_b = 0$ (airplane)

Figure 9: **Image backdoor.** Example of a backdoored image from the CIFAR10 dataset (Krizhevsky et al., 2009).

We assess whether a model has memorized backdoored image $I_b$ by prompting the model with $I_b$, and testing whether the produced label is $l_b$. In our experiments, we backdoor $Q = 10\%$ of a given device's training data, and we also backdoor $Q = 10\%$ of the global test data with which we evaluate all models in a topology.

**For the language data** (TinyMem) we create the OOD training data using the backdoor design introduced by Sakarvadia et al. (2025) which we formally define in Def. B.2. For all experiments, we let $t = 100$, $T = 2$.

**Definition B.2** (Language **Backdoor**). *Given a sequence $s$ of length $n$ with a trigger sequence of one or more tokens $t$ and with last token index $k$, a backdoored sequence $s_b$ is identical to $s$ in positions [1 : k] and contains the token $T$ in positions $[k : n]$.*

For example, if $t = [10]$, $T = 2$, $k = 5$, and $s = [2, 4, 6, 8, \mathbf{10}, 12, 14]$, then $s_b = [2, 4, 6, 8, \mathbf{10}, 2, 2]$.

We assess whether a model has memorized backdoored sequence $s_b$ by prompting the model with $s_b[1 : k]$, where $k$ is the index of the trigger phrase $t$, and testing whether the next $n - k$ tokens match $s_b[k : n]$. In our experiments, we partition $Q = 10\%$ of a given device's training data to be backdoored, we call this partitioned set $B$. We let the random trigger sequence be $t =$"100". Then, for any sequence in $B$ that contains $t$, we apply the backdooring procedure outlined in Def. B.2. We apply this backdooring procedure to $Q = 10\%$ of the global test data with which we evaluate all models in a topology.

### B.3 BASELINE AGGREGATION STRATEGIES

Here we provide the technical details of each baseline aggregation strategy we use to compare against our proposed solutions are: `Unweighted`, `Weighted`, `Random`, `FL`. We propose *topology-aware* aggregation strategies: `Degree`, `Betweenness`; details for topology-aware strategies are found in Section 4.

1. `Unweighted`: Given device $i, \forall j \in \mathcal{N}_i, C_{i,j} = \frac{1}{|\mathcal{N}_i|}$. (Devices weighted uniformly in a neighborhood.)

2. `Weighted`: Given device $i, \forall j \in \mathcal{N}_i, C_{i,j} = \frac{|train_j|}{\sum_{x \in \mathcal{N}_i} |train_x|}$ where $train_x$ is the training dataset for device $x$. (Devices weighted by their respective training dataset size in a neighborhood.)

3. `Random`: Given device $i, \forall j \in \mathcal{N}_i, C_{i,j} = \frac{e^{\frac{R_j}{\tau}}}{\sum_{k \in \mathcal{N}_i} e^{\frac{R_k}{\tau}}}$ where $R \in \mathbb{R}^{|\mathcal{N}_i|}$ is uniformly sampled random vector. (Devices weighted by uniformly random coefficient which are scaled by a softmax with temperature $\tau$.)

4. `Federated Learning (FL)`: Given device $i, \forall j \in \mathcal{T}, C_{i,j} = \frac{1}{|\mathcal{T}|}$ where $\mathcal{T}$ is the topology. (Simple federated learning baseline.)

### B.4 TOPOLOGY-AWARE VS. UNAWARE

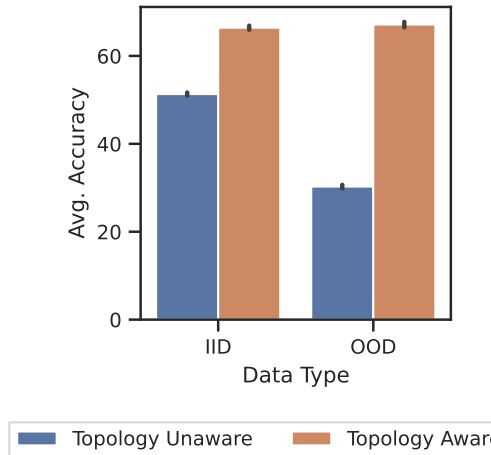

Figure 10: **Topology-Aware Aggregation strategies outperform Topology-unaware strategies.** Average test accuracy for IID vs. OOD knowledge averaged across all models in a topology after $R = 40$ rounds of training. Averaged across 3 **Barabási-Albert** topologies with preferential attachment parameter $p \in \{1, 2, 3\}$, five datasets (MNIST, FMNIST, CIFAR10, CIFAR100, TinyMem), four different OOD data locations (varied across top four nodes with highest degree in a given topology), and three seeds. Topology-aware aggregation strategies are `Degree` and `Betweenness`. Topology-unaware aggregation strategies are `Unweighted`, `Weighted`, `Random`, and `FL`.

# C  COMPUTATIONAL COST OF DECENTRALIZED LEARNING

Here we break down the three costs associated with decentralized learning algorithms: upfront costs, memory costs, and communication costs. Topology-aware costs incur *identical* communication and memory costs to topology-unaware baselines (see Table 2). Further, topology-aware algorithms have a negligible one-time cost of computing centrality metrics which amortizes over time (see Section C.1).

| Method | Upfront Cost (time) | Memory | Communication (at each round) |
|---|---|---|---|
| Degree | $O(1)$ | | |
| Betweenness | $O(nm + n2logn)$ | $O(|M|)$ | $O(|N|)$ |
| Random | time to compute $n$ random numbers | $O(|M|)$ | $O(|N|)$ |
| Weighted | time to count data samples per node | $O(|M|)$ | $O(|N|)$ |
| Unweighted | - | $O(|M|)$ | $O(|N|)$ |
| FL* | - | $O(|M|)$ | $O(1)$ |

Table 2: Upfront, memory, and communication cost complexity analysis for an individual device in a topology. $n$ and $m$ are the number of nodes and edges, respectively, in the topology. $|M|$ indicates size of a model. $|N|$ indicates the number of neighbors that node has. *FL is not truly a *decentralize* technique.

## C.1  UPFRONT COST: COMPUTING CENTRALITY METRICS

We study both Degree and Betweenness. We profile the cost of computing both here.

The time complexity of computing Degree is $O(1)$. In undirected graphs, the setting we study, the time complexity of calculating Betweenness is $O(nm + n2logn)$ where $n$ and $m$ are the number of nodes and edges respectively using Brandee's algorithm (Brandes, 2001). We profile the cost of this algorithm and note that for graphs < 1000 nodes the cost is less than 1 second, and for 1024 nodes the cost is 2.35 seconds (See Table 3). We also include cost of computing Closeness and Degree as a comparison.

| Number of Nodes | Betweenness (s) | Closeness (s) | Degree (s) |
|---|---|---|---|
| 8 | 0.00024 | 0.048 | 0.000008 |
| 16 | 0.0057 | 0.00016 | 0.000008 |
| 32 | 0.013 | 0.0005 | 0.000011 |
| 64 | 0.025 | 0.00017 | 0.000017 |
| 128 | 0.12 | 0.0062 | 0.000039 |
| 256 | 0.24 | 0.024 | 0.000048 |
| 512 | 0.68 | 0.099 | 0.000096 |
| 1024 | 2.35 | 0.41 | 0.000203 |
| 10,000 | 380 | 43 | 0.0018 |

Table 3: Time it takes to compute Betweenness, Degree and Closeness for topologies of varying node counts on CPU.

## D   THEORY

The intuition for topology-aware metrics is drawn from the field of information diffusion/opinion maximization in social networks (Gionis et al., 2013), whereby information can be routed faster through a social network topology via topology-aware algorithms. The extension of these concepts to decentralized learning is natural as topology can similarly be accounted for during aggregation steps.

Because our decentralized learning problem is posed in a traditional gossip learning framework (but with topology-aware mixing coefficients), previously established convergence bounds for non-convex, non-IID optimization apply (Koloskova et al., 2020).

To provide theoretical justification for why we expect topology-aware aggregation coefficients to enhance information spread, we do a spectral gap analysis. A spectral gap of a matrix is the difference between its two largest eigenvalues. Decentralized learning theory has established that a larger spectral gap is beneficial to learning. Vogels et al. (2022) write "the graph topology appears through the spectral gap of its averaging (gossip) matrix. The spectral gap poses a conservative lower bound on how much one averaging step brings all workers' models closer together. The larger, the better."

We analyze the spectral gap of the averaging (gossip) matrix of several of the graphs we study under varying aggregation schemes. We choose to study **Barabási-Albert** (BA) graphs with n=33 nodes and varying degree parameter p (a common model of real world topologies – see Fig 11). We include a complete graph n=33. In all cases, the topology-aware methods (`Degree`, `Betweenness`) achieve a higher spectral gap than topology-unaware counterparts which may explain their success at propagating knowledge (See Table 4).

| Topology | Betweenness $\tau =$ 0.1 | Degree $\tau =$ 0.1 | Unweighted | Random $\tau =$ 0.1 |
|---|---|---|---|---|
| **BA** $p = 1$ (low degree) | **0.29** | **0.029** | 0.013 | 5.14e-7 |
| **BA** $p = 2$ (med. degree) | **0.35** | **0.32** | 0.17 | 2.18e-2 |
| **BA** $p = 3$ (high degree) | **0.47** | **0.55** | 0.34 | 0.10 |
| Complete | 1.02 | 1.02 | 1.02 | 1.00 |

Table 4: Spectral gap analysis of topology-aware (`Degree` and `Betweenness`) versus topology-unaware (`Unweighted`, `Random`) aggregation strategies. Higher spectral gap is beneficial.

# E    TOPOLOGIES

## E.1    STUDIED TOPOLOGIES

We visualize all the topologies that we study in our experiments in Figs 11–14.

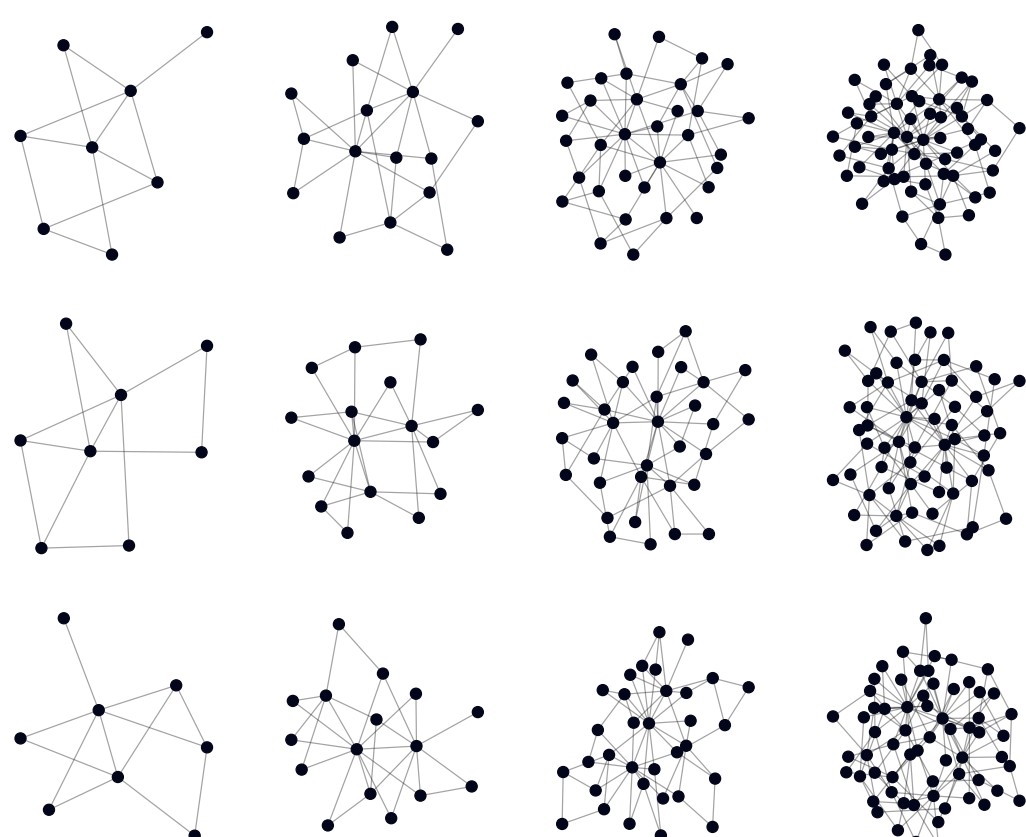

Figure 11: **Barabási-Albert topologies.** Preferential attachment parameter $p = 2$ for all topologies. Left to right: topologies have n $\in \{8, 16, 33, 64\}$ nodes. Top to bottom: seeds $\in \{0, 1, 2\}$.

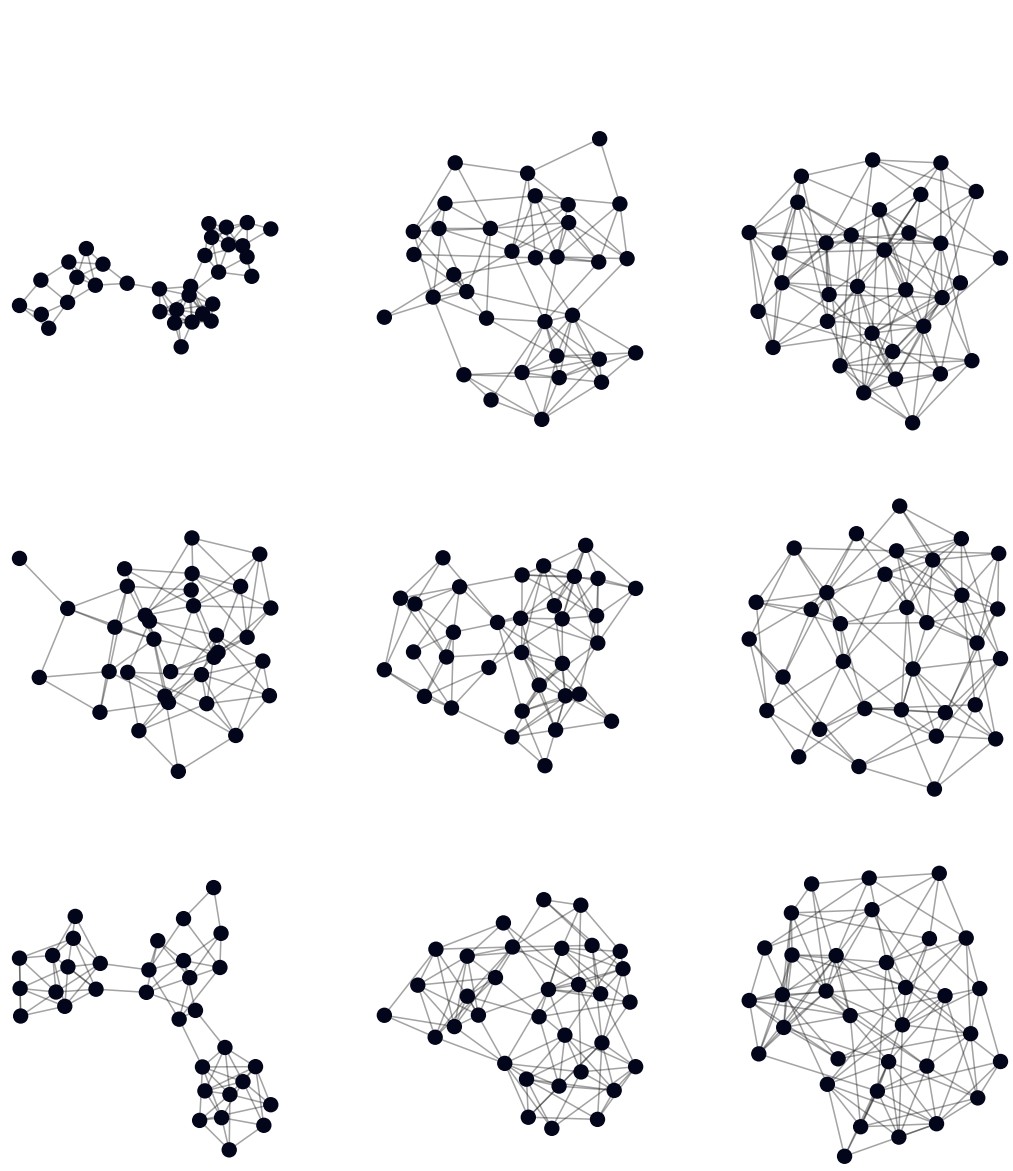

Figure 12: **Stochastic Block topologies.** Each topology has three communities with varying levels of modularity. Each topology has $n = 33$ nodes. Left to right: the probabilities of edges existing between communities $m_i$ to $m_j$ are $p_{i,j}$: if $i = j, p_{i_j} = 0.5$, if $i \neq j$, then we varied $p_{i_j} \in \{0.009, 0.05, 0.9\}$. Top to bottom: seeds $\in \{0, 1, 2\}$.

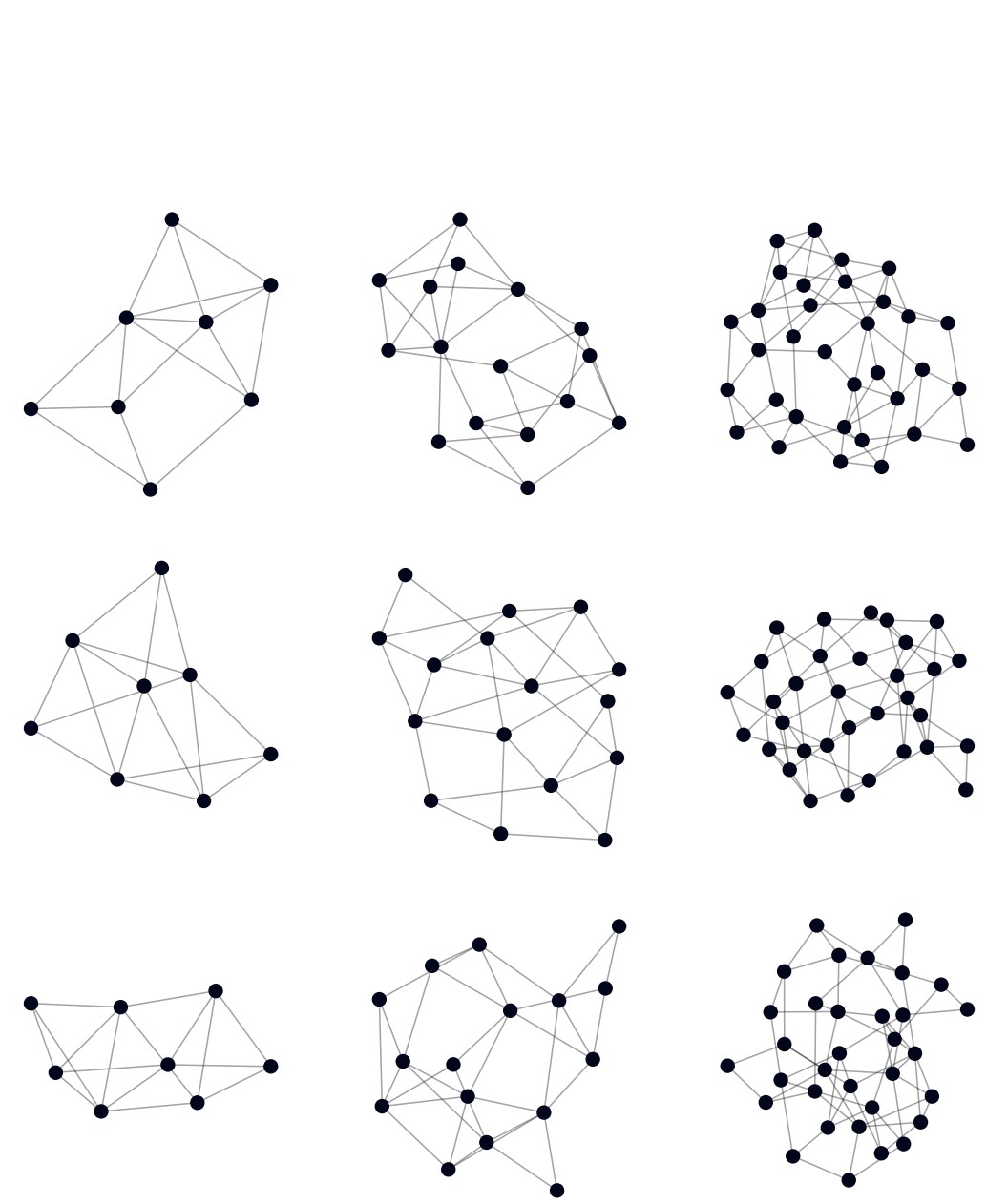

Figure 13: **Watts-Strogatz topologies.** Each topology has $k = 4$, $u = 0.5$. Left to right: topologies have n $\in \{8, 16, 33\}$ nodes. Top to bottom: seeds $\in \{0, 1, 2\}$.

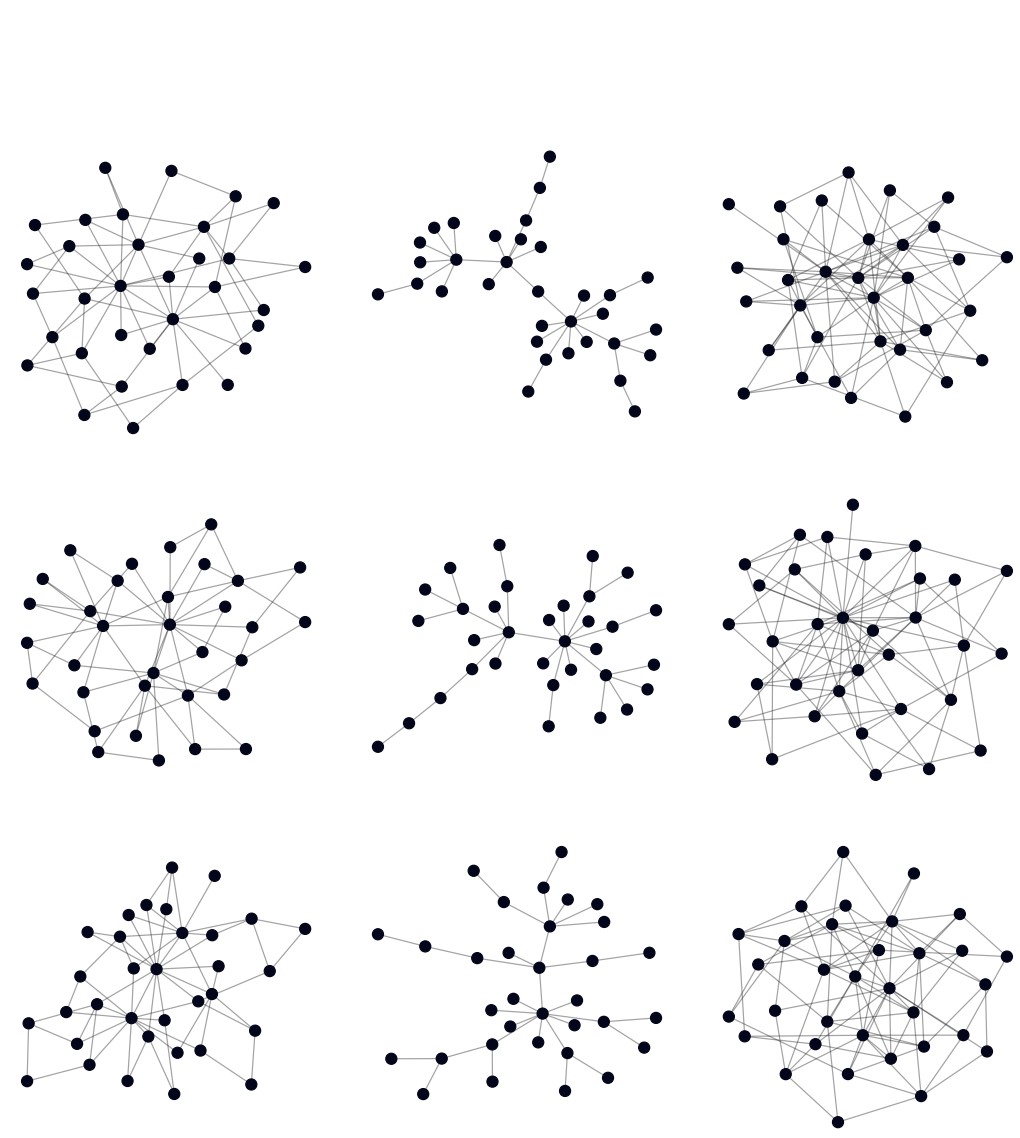

Figure 14: **Barabási-Albert topologies.** Each topology has $n = 33$ nodes. Left to right: topologies have preferential attachment parameter $p \in \{2, 1, 3\}$. Top to bottom: seeds $\in \{0, 1, 2\}$.

## E.2 IMPACT OF TOPOLOGY

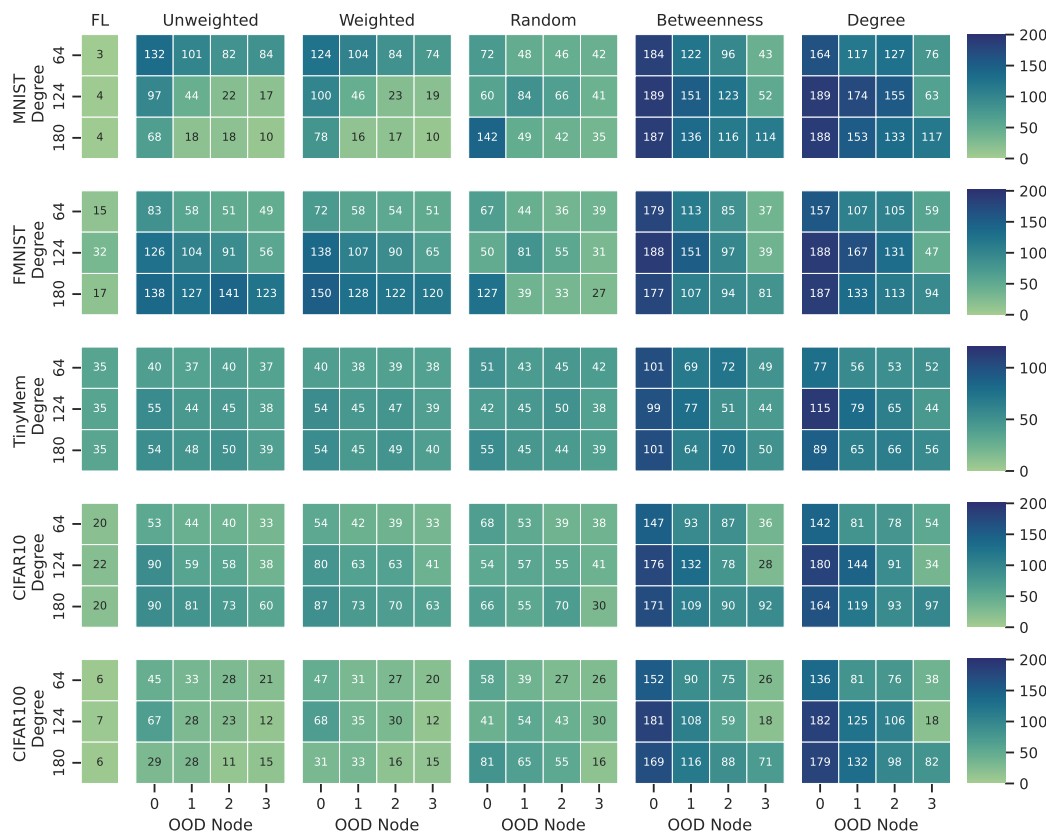

Figure 15: **Impact of topology degree on aggregation strategy performance.** Experiments performed across **Barabási-Albert** topologies with $n = 33$ nodes and $p \in \{1, 2, 3\}$. Higher $p$ means higher degree. Results averaged across three seeds.

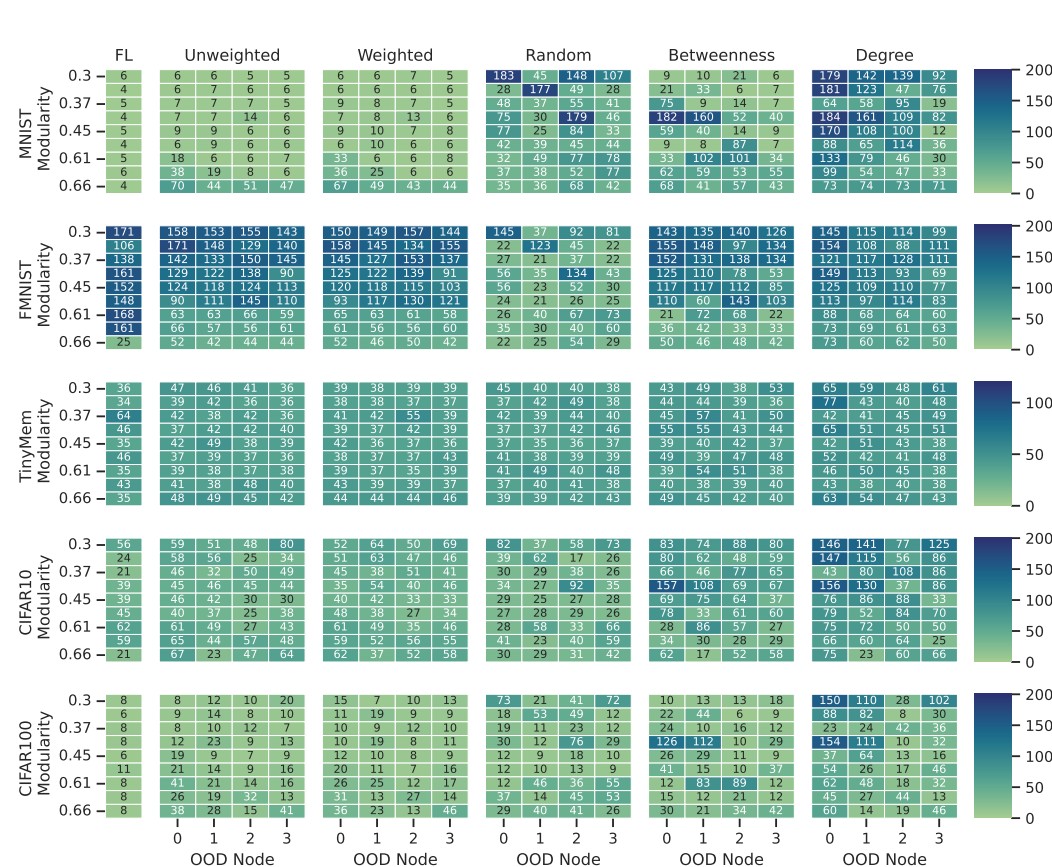

Figure 16: **Impact of modularity on aggregation strategy performance.** Experiments performed across **Stochastic Block** topologies with $n = 33$ nodes and three communities with varying levels of modularity. The probabilities of edges existing between communities $m_i$ to $m_j$ are $p_{i,j}$: if $i = j$, $p_{i_j}$ = 0.5, if $i \neq j$, then we varied $p_{i_j} \in \{0.009, 0.05, 0.9\}$. Results shown for three seeds. (As a newly seeded **SB** is generated, its "modularity" score changes from the previous seed; therefore, we do not average across seeds in this figure.)

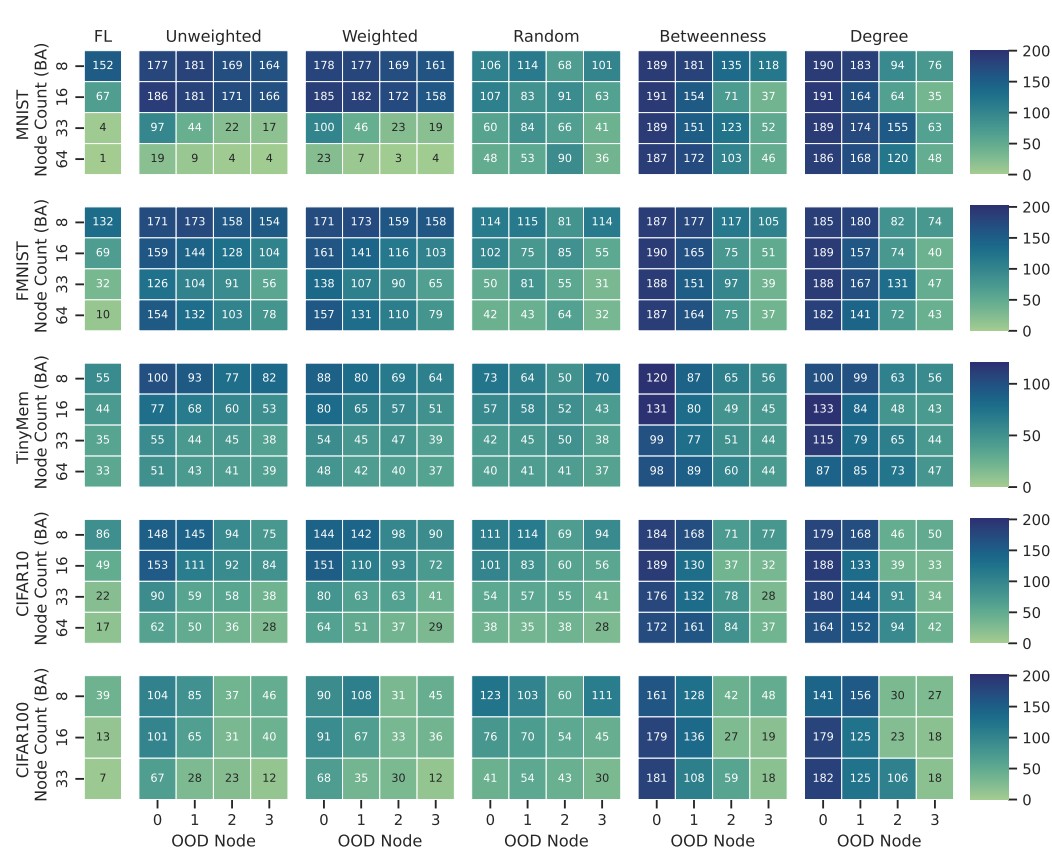

Figure 17: **Impact of topology node count on aggregation strategy performance.** Experiments performed across **Barabási-Albert** topologies with $p = 2$, $n \in \{8, 16, 33, 64\}$ nodes. Results averaged across three seeds. We exclude experiment for CIFAR100 on **BA** topologies w/ $n = 64$ due to computational cost.

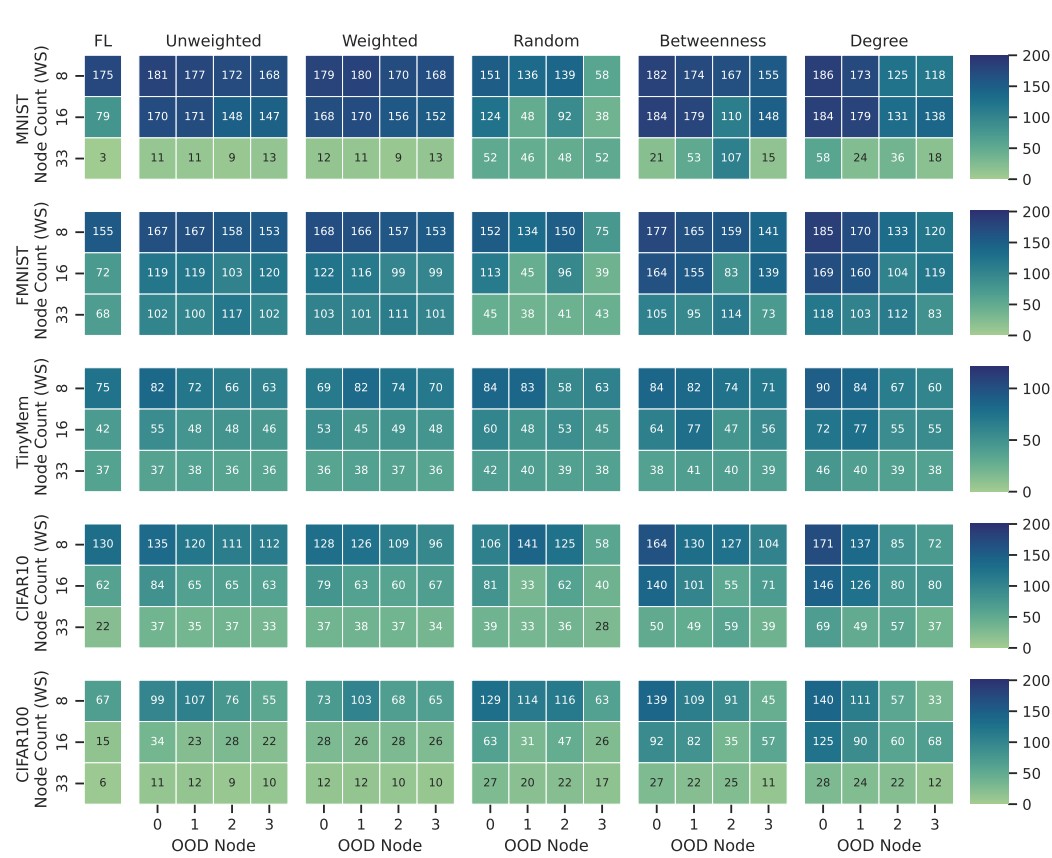

Figure 18: **Impact of topology node count on aggregation strategy performance.** Experiments performed across **WS** topologies with $k = 4$, $u = 0.5$, and $n \in \{8, 16, 33, 64\}$ nodes. Results averaged across three seeds.

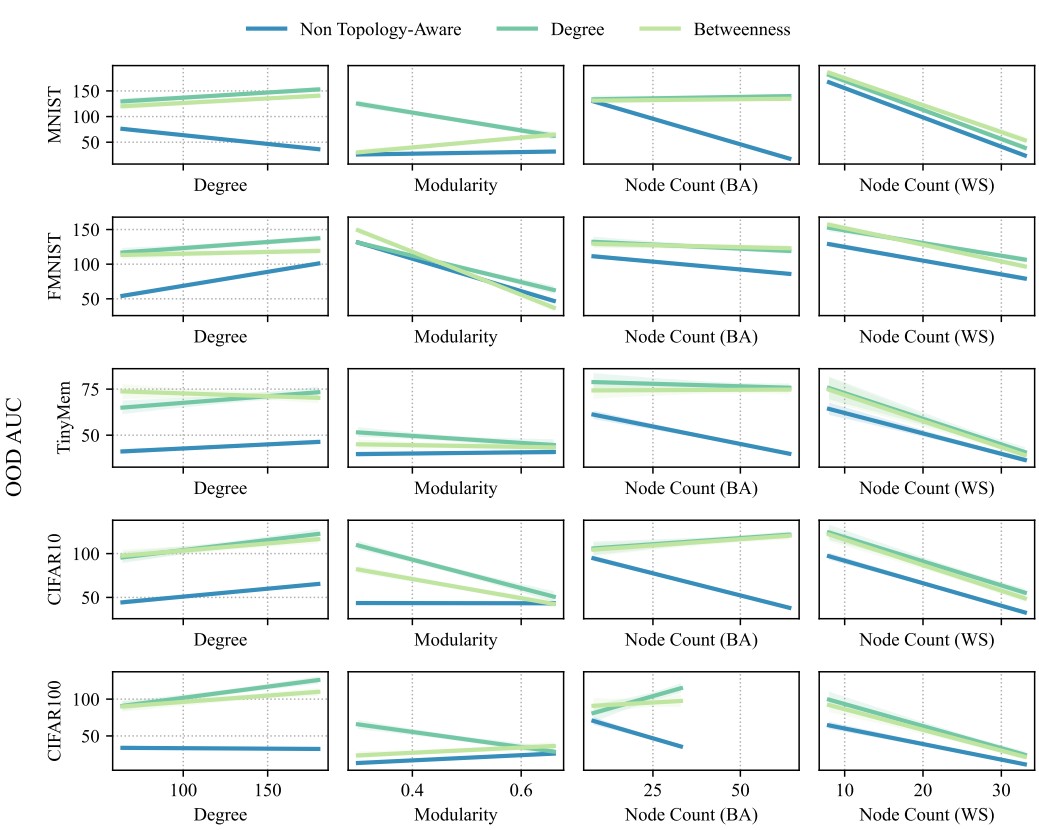

Figure 19: **Impact of topology degree, modularity, node count on aggregation strategy performance.** From left to right: we plot the impact of topology degree, modularity, and node count on the OOD test accuracy AUC. Higher is better (indicates higher propagation of OOD knowledge). We exclude experiment for CIFAR100 on **BA** topologies w/ $n = 64$ due to computational cost.

### E.3 BARABÁSI-ALBERT VS. WATTS-STROGATZ DEGREE DISTRIBUTIONS

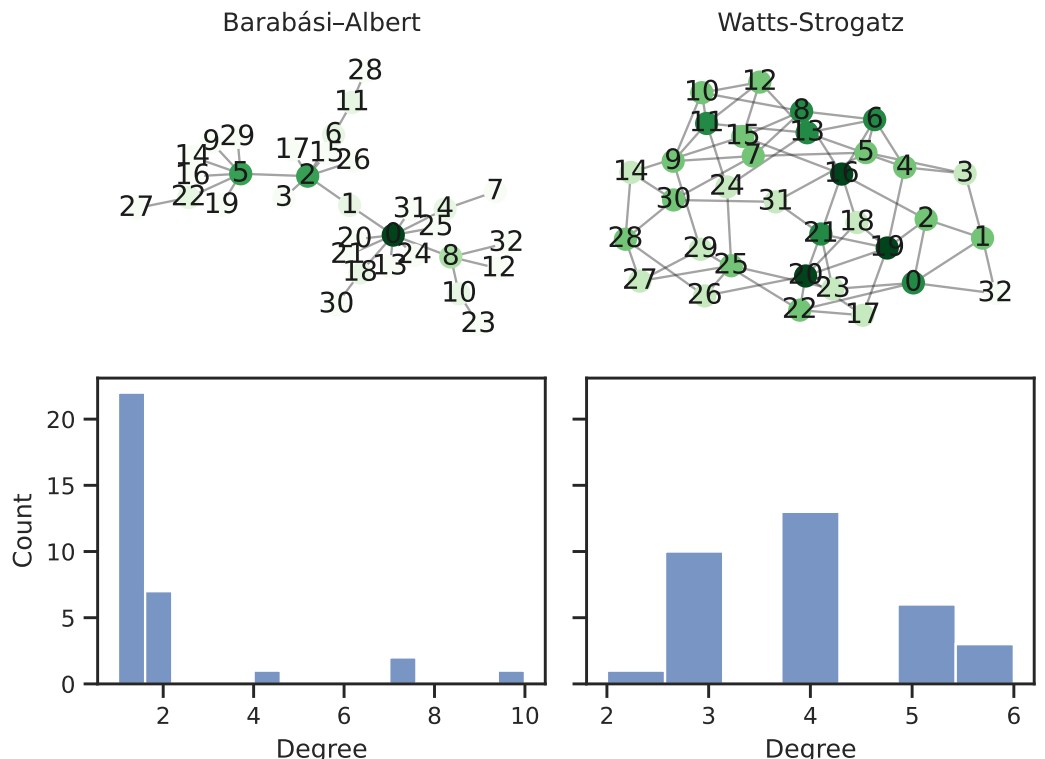

Figure 20: **Barabási-Albert vs. Watts-Strogatz degree distributions.** Histograms show the relative degree distributions show the relative degree distributions of both the **BA** and **WS**. Both topologies have $n = 33$ nodes. **BA** has $p = 1$. **WS** has $k = 4$, $u = 0.5$. **BA** has a power-law degree distribution across its nodes while the **WS** has a more normal degree distribution. Numbers on nodes indicate node identification number. Nodes colored by degree count (darker means higher degree).

Table 5: **System parameters** for Aurora at ALCF (ALCF, 2025) to estimate energy and carbon usage.

| Machine | $c_{\text{uf}}$ | $c_{\text{tdp}}$ | ngpu | $g_{\text{uf}}$ | $gpu_{tdp}$ | $DRAM$ | $PUE$ |
|---------|------|------|------|------|------|------|------|
| Aurora | 0.5 | 350 | 6 | 1 | 600 | 1024 | 1.58 |

Table 6: **Experiment counts** across topologies with varying device counts.

| Data | 8 Devices | 16 Devices | 33 Devices | 64 Devices |
|------|-----------|------------|------------|------------|
| MNIST | 126 | 126 | 441 | 63 |
| FMNIST | 126 | 126 | 441 | 63 |
| TinyMem | 126 | 126 | 441 | 63 |
| CIFAR10 | 126 | 126 | 441 | 63 |
| CIFAR100 | 126 | 126 | 441 | n/a |

## F LIMITATIONS

Our two proposed topology-aware aggregation strategies (Degree, Betweenness) only perform substantially differently from Unweighted in topologies in which node-level topology characteristics vary substantially between nodes. Further, topology-aware aggregation strategies can propagate all types of data, including both benign and malicious data; therefore, it is important to have safeguards to detect and remove unwanted training data.

## G COMPUTATION RESOURCES AND ENERGY USAGE

The experiments in this paper used approximately **70,178 kWh of energy, and 21,0153 Kg of carbon**.

We detail the computational resource (i.e., node hours, energy, carbon) used by our experiments below.

To calculate node hours, we calculate the average time of all our final experiments for two aggregation rounds across three trials. We then multiple by 20 for each experiment as we run each experiment for 40 aggregation rounds. We then *triple that value* to account for extra debugging time, faulty runs, and any unaccounted for compute usage (see Table 8). All experiments were run on the Aurora machine at Argonne Leadership Computing Facility (ALCF, 2025).

To calculate energy usage and carbon cost of our experiments, we follow the methodology detailed by Bouza et al. (2023):

$$energy = NH * ((c_{\text{uf}} * c_{\text{tdp}}) + (ngpu * g_{\text{uf}} * gpu_{\text{tdp}}) + (0.3725 \text{ W/Gb} * DRAM)) * PUE, \quad (3)$$

where *NH* is node hours, $c_{\text{uf}}$ is the CPU usage factor, $c_{\text{tdp}}$ is the CPU's thermal design power, *ngpu* is the number of GPUs on a node, $g_{\text{uf}}$ is the GPU usage factor (we assume 100% utilization), $g_{\text{tdp}}$ is the GPU's thermal design power, *DRAM* is dynamic random access memory, and *PUE* is the power usage efficiency. *energy* is reported in watt hours. We record system-specific parameter values in Table 5.

$$carbon = (energy/1000) * CI \quad (4)$$

Above in Eq (4), *energy* is obtained in watt hours from Eq (3), and *CI* is the carbon intensity reported based on the yearly regional average for each computing center from ElectricityMaps (2025). For Aurora's geographic location in 2025, *CI* = 300 g/kWh. *carbon* is reported in grams.

Table 7: **Energy and carbon estimates** for our experiments based on Eq (3) & Eq (4) respectively.

| Data | Machine | Node Hours | Energy (kWh) | Carbon (Kg) |
| --- | --- | --- | --- | --- |
| MNIST | Aurora | 1229.12 | 8071.85 | 2421.55 |
| FMNIST | Aurora | 1186.30 | 7790.65 | 2337.20 |
| TinyMem | Aurora | 2336.25 | 15342.57 | 4602.77 |
| CIFAR10 | Aurora | 3679.14 | 24161.54 | 7248.46 |
| CIFAR100 | Aurora | 2255.33 | 4443.35 | 4443.35 |
| All Experiments (total) | – | 10686.14 | 70177.78 | 21053.33 |

Table 8: **Single Experiment Node hours.** Estimates from **BA** topology with $p = 3$ for 40 rounds of training. We exclude experiment for CIFAR100 with 64 device topologies due to computational constraints. All times tripled to account for extra debugging time, faulty runs, and any unaccounted for compute usage.

| Data | 8 Devices | 16 Devices | 33 Devices | 64 Devices |
| --- | --- | --- | --- | --- |
| MNIST | 0.97 | 1.10 | 1.74 | 3.20 |
| FMNIST | 0.76 | 1.07 | 1.71 | 3.19 |
| TinyMem | 1.62 | 2.06 | 3.39 | 6.01 |
| CIFAR10 | 2.33 | 3.15 | 5.37 | 9.84 |
| CIFAR100 | 2.34 | 2.65 | 3.69 | n/a |

# H LABEL HETEROGENEITY

Here we assess the impact of topology-aware vs. topology-unaware algorithms for heterogeneous data distributions characterized by label skew. We perform an experiment were we vary $\alpha_l \in \{1, 10, 1000\}$ (high to low label heterogeneity) and we assess how effective different aggregation methods are at propagating knowledge. Noteable, in these experiments we do not place a single source of OOD data on a specific node. Instead, by inducing label heterogeneity, data distributions across nodes are out-of-distribution with respect to each other to different degrees. We find that at all levels of heterogeneity, topology-aware methods **outperform** their topology-unaware counterparts.

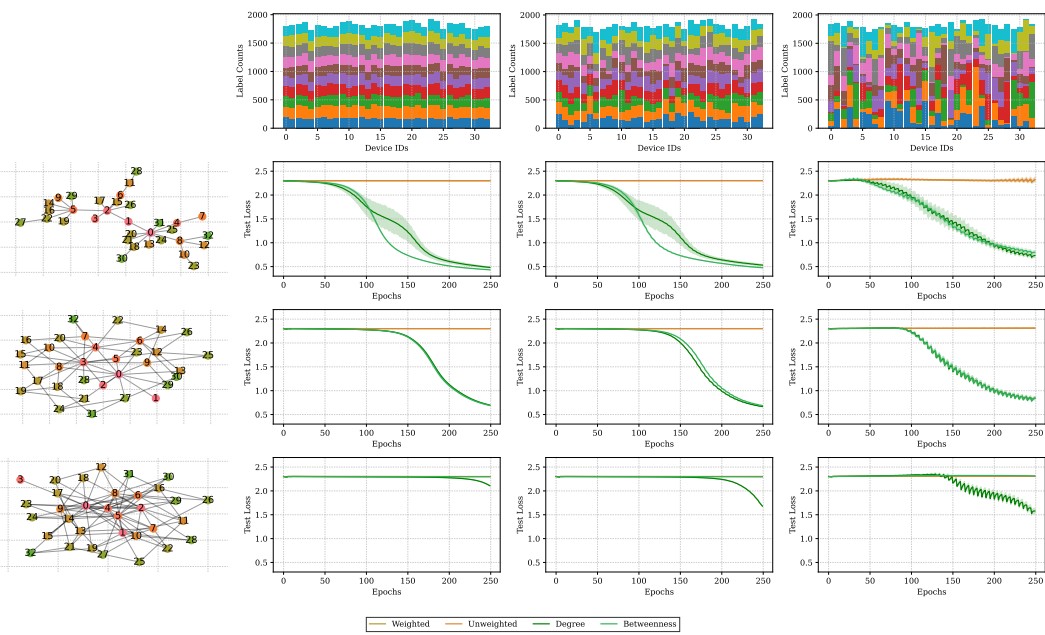

Figure 21: **Label Heterogeneity.** In each case, topology-aware methods (`Degree`, `Betweenness`) are more successful at propagating knowledge than topology-unaware methods. Left to right: $\alpha_l \in \{1000, 10, 1\}$. Top row visualizes the distribution of label classes across devices. We vary the aggregation strategies: `Weighted`, `Unweighted`, `Betweenness` ($\tau$=0.1), `Degree` ($\tau$=0.1). Lower test loss is better.

## I TEMPERATURE SENSITIVITY

Here we tune $\tau$, the temperature of the softmax used to normalize the topology aware coefficients. For MNIST on a **Barabási-Albert** graph (w/ 33 nodes and p=1), $(\alpha_s, \alpha_l) = (1000,1000)$, and OOD data placement is varied across the top five nodes with highest degree. We notice that $\tau = 0.1$ has a reasonable performance tradeoff between OOD AUC and IID AUC (see Fig 22).

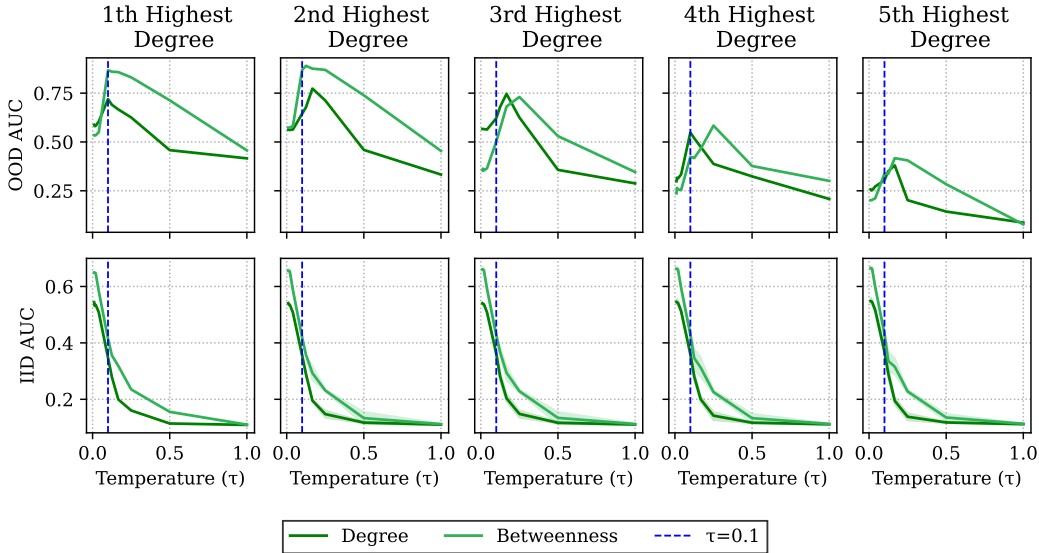

Figure 22: **Tuning Temperature Parameter.** Left to right: OOD data placement is varied across the top five highest degree nodes. We vary the aggregation strategies: `Betweenness`, `Degree`. Top row reflects AUC of OOD test set averaged across all devices. Bottom row reflects AUC of IID test set averaged across all devices. Higher AUC is better.

