# OpenReview forum: "Topology-Aware Knowledge Propagation in Decentralized Learning"
_ICLR.cc/2026/Conference — Submitted to ICLR 2026_

### Official Review · Reviewer_ggNV · 2025-10-25

**Soundness:** 3
**Presentation:** 2
**Contribution:** 2
**Rating:** 2
**Confidence:** 3

**Summary:**

This paper empirically studies the impact of the topology of network, such as the degree of nodes, the number of nodes, and modularity, on the propagation of out-of-distribution (OOD) knowledge in distributed model training, finds that existing model aggregation methods can not propagate OOD knowledge effectively and proposes a topology-aware aggregate framework for enhancing the of propagation of OOD knowledge. The effectiveness of the proposed aggregation strategies are verified by extensive experiments.

**Strengths:**

* It is an important question that how network topology impacts the performance of distributed model training algorithms.

* This paper proposes new aggregation algorithms by incorporating the topology information, such as degree and betweenness centrality.

**Weaknesses:**

* The empirical findings in this paper is incremental in comparison with previous ones in [1]. It is better to provide some theoretical results.

* There is a lack of proper definition regarding the performance measurement of algorithms (or the measure for knowledge propagation). In line 184, this paper uses the average AUC of all models in a topology on a given test set to measure the knowledge propagation. It is not a proper measurement. Since each device maintains its own local model, it is better to measure the test performance of each model on its own test data.

* There is a lack of theoretical analysis on the generalization error bound of the proposed algorithms. It is intuitive that the generalization error bound should depend on the topology of network, such as the connectivity of network, the degree of nodes and the location of OOD datasets.

References

[1] Palmieri et al. Impact of network topology on the performance of decentralized federated learning. CoRR, 2024.

**Questions:**

Please refer to the above-mentioned weaknesses.

---

> ### Author Response · Authors · 2025-11-19
>
> Thank you for your review. We are glad you find our work to tackle an important problem.
> - **W1**: Previous work is an analysis that determines that topology plays a role in knowledge propagation during decentralized learning. A key finding in [1] is “when the highest degree nodes enjoy higher accuracy, they are able to drag all the other nodes closer to their performance efficiently”. We significantly build upon and expand the findings of [1] by proposing to **utilize knowledge of topology to enhance** knowledge propagation via topology-aware aggregation schemes.
> - **W2**: We disagree with the statement that “it is better to measure the test performance of each model on its own test data”. In decentralized learning, we are interested in **global** knowledge propagation **across** devices in the topology. To do this, we need a metric that can clearly track how knowledge is spreading across nodes – by only considering test-device-specific assessment, you fail to account for knowledge spread and cannot assess the local models on out-of-distribution data. Instead, the metric we propose, the area under the curve of model accuracy on a global held out test set is more appropriate. First, a global test set can reflect the characteristics of the global data distribution across all devices, and therefore can reflect whether a model has generalized **beyond** its local training distribution. Second, by measuring the AUC, rather than final model accuracy, we can account for how **quickly** knowledge was propagated and acquired over training in addition to the final accuracy. The AUC allows us to summarize the extent to which knowledge has spread on average across a moving time horizon (it gives you a strong notion of how fast knowledge is spreading).
> - **W3**: We already include a theoretical justification of our methodology in Appendix D. We summarize here: Because our decentralized learning problem is posed as a traditional gossip learning framework (but with topology-aware mixing coefficients), previously established convergence bounds for non-convex, non-IID optimization apply [1]. To provide theoretical justification for why we expect topology-aware aggregation coefficients to enhance information spread, we do a spectral gap analysis. A spectral gap of a matrix is the difference between its two largest eigenvalues. Decentralized learning theory has established that “the graph topology appears through the spectral gap of its averaging (gossip) matrix. The spectral gap poses a conservative lower bound on how much one averaging step brings all workers’ models closer together. The larger, the better. [2]” Therefore to motivate our work from a theoretical perspective, we analyze the spectral gap of the averaging (gossip) matrix of several of the graphs we study under varying aggregation schemes. In all cases, the topology-aware methods (Degree, Betweenness) achieve a higher spectral gap than topology-unaware counterparts which may explain their success at propagating knowledge (See Table 4, Appendix D).
>
> [1] A Unified Theory of Decentralized SGD with Changing Topology and Local Updates, ICML, 2020.
>
> [2] Beyond spectral gap: The role of the topology in decentralized learning, NEURIPS, 2022.

---

> > ### Comment · Reviewer_ggNV · 2025-11-23
> > **I thank the authors for detailed rebuttal that have addressed my second concern.  The other concerns remain unchanged.**
> >
> > In terms of the third concern, Appendix D did not establish a proper generalization analysis.
> > It is better to give an explicit generalization bound and the correspinding proof.
> >
> > (1) It was claimed that the proposed decentralized learning problem can be posed in a traditional gossip learning framework [1].
> >
> > First, it is necessary to establish a rigorous reduction between the two problems.
> > Second, although previous analysis in [1] is applicable to the problem in this paper,
> > a concise proof and the corresponding theortetical bound should still be clearly stated for the completeness and correctness.
> >
> > (2) It is more important to show that how the proposed algorithm with topology-aware mixing coefficients improves the generalization error bound compared to prior algorithms.
> >
> >
> > References
> >
> > [1] Koloskova et al. A unified theory of decentralized SGD with changing topology and local updates. ICML, 2020.

---

### Official Review · Reviewer_jFih · 2025-10-27

**Soundness:** 3
**Presentation:** 3
**Contribution:** 2
**Rating:** 4
**Confidence:** 4

**Summary:**

This paper tackles an important and challenging problem in decentralized learning: the effective propagation of out-of-distribution (OOD) knowledge across a network topology. The authors argue that traditional aggregation strategies, being unaware of the network structure, hinder OOD knowledge propagation, making performance highly dependent on the location of the OOD data source. To address this, the paper introduces "topology-aware aggregation strategies", which leverage network centrality metrics like node degree and betweenness to weight the model aggregation process, thereby accelerating the diffusion of critical knowledge. Through extensive experiments across various topologies and datasets, the authors demonstrate that their proposed methods significantly outperform existing topology-unaware baselines in improving OOD accuracy.

**Strengths:**

Strong Motivation:The problem of OOD knowledge propagation is highly practical and critical in real-world decentralized learning scenarios, such as IoT and edge computing.

Comprehensive Experiments:The experimental setup is very extensive, covering a wide range of variables and providing strong empirical evidence for the conclusions.

Theoretical Support:The spectral gap analysis in the appendix offers a plausible theoretical explanation for why the topology-aware strategies are superior, moving beyond purely empirical findings.

**Weaknesses:**

There is a lack of clear definitions for OOD and IID knowledge in the decentralized context, which obscures the relationship and distinction between them.

The paper provides insufficient justification for selecting degree and betweenness centrality over other potential topological metrics, leaving it unclear whether these are the optimal choices.

The analysis is confined to static topologies; applicability and potential overhead in dynamic networks are not discussed.

**Questions:**

There are some questions as follows：

1.Could you provide clearer definitions for IID and OOD knowledge in the decentralized context to better distinguish them?

2.What was the rationale for choosing degree and betweenness centrality over other metrics, and is there evidence they are optimal?

3.How would your method perform in dynamic topologies, and what is the expected overhead from re-calculating centrality?

4.Given that your topology-aware approach intentionally amplifies the influence of central devices, how do you propose to mitigate the heightened security risk of a compromised central device rapidly spreading malicious knowledge throughout the network?

5.Could you elaborate on the relationship between your specific OOD setup and more general Non-IID settings, like label skew across all devices?

6.Regarding contribution two, you state the sensitivity to topology is "a problem which does not exist in FL". What "problem" does this refer to, and why is it absent in FL?

7.How would your method's performance change if all devices had varying degrees of Non-IID data, instead of the current single-source OOD setting?

---

> ### Author Response · Authors · 2025-11-19
>
> Thank you for your review. We are glad you find our work to have strong motivation, and comprehensive experiments.
> - **W1/Q1 IID vs. OOD**: We clarify the manner in which we characterize our data distributions across devices in Appendix B.2.1 (IID data) and B.2.2 (OOD Data). To summarize, there are two key parameters we use to control the heterogeneity of data across devices: αₛ (distribution of sample counts),  αₗ (distribution of label counts). As  (αₛ, αₗ) go to infinity data distributions become more homogeneous across both axis, as  (αₛ, αₗ) go to 0 data distributions become more heterogeneous (see Figure 8). We note that there is no strict convention for what warrants certain data to be OOD w.r.t. other data; therefore, we formulate experiments in a manner that would allow us to track the spread of data carefully. We do this by first distributing data IID across devices, and then picking one device on which to convert 10% of the data to OOD (as outlined in B.2.2); this enables us to carefully track this OOD set as it is learned by other devices over training.
> - **W2/Q2 Choice of centrality metrics**: Good question! We discuss additional centrality metrics on lines 239-245. We choose degree and Betweenness as they capture local and global phenomena, respectively, and together expose a wide spectrum of knowledge propagation behavior. As this is the first exploratory study into topology-aware aggregation, we leave analysis of additional metrics to future work.  We instead prioritize a comprehensive study of both Degree/Betweenness. Through our experiments, Betweenness and Degree metrics of individual nodes do not substantially differ from one another. This is empirically validated in our work (see Figures 4, 6), where we observe very similar knowledge propagation performance between degree and Betweenness; in both cases we substantially outperform topology-unaware baselines. Therefore, we posit that our choice of centrality metrics for this initial work, under realistic real-world topology choices, is sufficient for arguing the necessity for topology-aware aggregation strategies.
> - **W3/Q3 Negligible overhead in Dynamic Topologies**: The cost of computing degree is O(1) and therefore negligible in both static and dynamic topologies. We detail the overhead of dynamically computing centrality metrics in Appendix C. Specifically in appendix C1. We profile the cost of computing betweenness and note that for graphs < 1000 nodes the cost is less than 1 second, and for 1024 nodes the cost is 2.35 seconds; **relative to the cost of training/communicating updates between nodes, this is negligible**. In static graphs, this is a **one time cost** incurred at the beginning of training which amortizes out over the course of training. The majority of time spent during training is for on-device training (detailed in Appendix D: Tables 4, 5). In the event that the topology is dynamically changing, **this cost would be incurred once** every time the topology changes. Given that Betweenness and Degree yield similar performance in real-world graphs (Figure 4,6), cheaper metrics like Degree should be used if the cost of Betweenness is prohibitive.
> - **Q4 Malicious actors**: The spread of corrupted or adversarial data is indeed a concern of **all** decentralized (and centralized) learning settings as we note in lines Appendix F in our limitations section. **This is not a limitation of our work, but rather an important consideration when designing any model training workflow.** Any robust training workflow needs to account for the possibility of malicious actors when collecting their training data. As with any machine learning setting, it is important to have safeguards to detect and remove unwanted training data. There are many approaches to detecting and spreading the spread of corrupted/malicious data. However, this study is the first of its kind: a large-scale empirical analysis of knowledge propagation in realistic real-world topologies. Effectively propagating knowledge in the first place remains an open problem (especially in the OOD setting). Accounting for corrupted/adversarial training data is important and outside scope of this investigation.
> - **Q6**: In FL the topology is **not between clients in a peer-to-peer manner.** Rather, the topology is just a star topology where clients are directly connected to a central aggregation server. Thus, every device receives the exact same update from the parameter service during each aggregation step. Therefore, the locations of individual client devices are **indistinguishable** from their neighbors (this means topology plays no role w.r.t. Individual devices in the topology).

---

> > ### Author Response · Authors · 2025-11-19
> >
> > - **Q5/Q7 More general Non-IID/more non-IID sources**: First, we emphasize that our initial experimental design with a single controlled source of OOD information enabled us to exactly track how knowledge flows across devices. Having more than one source of OOD data makes it hard to distinguish the impact of topology vs. the impact of data on knowledge spread. Despite this, we run additional experiments to clarify the effect of more realistic non-IID data heterogeneity found in real-world settings by varying αₗ (distribution of label counts). As (αₗ) goes to infinity, data distributions become more homogeneous; in this setting, there are multiple sources of OOD data (varies depending on αₗ). Please see **Appendix H, Figure 21** for full experiment details and results. **In summary, we find that at all levels of heterogeneity (multiple sources of OOD data), topology-aware methods outperform their topology-unaware counterparts.**

---

### Official Review · Reviewer_Esqg · 2025-10-28

**Soundness:** 2
**Presentation:** 3
**Contribution:** 3
**Rating:** 2
**Confidence:** 3

**Summary:**

This work studies the knowledge propagation in decentralized learning. In particular, the authors find that popular algorithms struggle to propagate out-of-distribution (OOD) knowledge to all devices. They further propose two topology-aware aggregation strategies to solve the problem. The experiments demonstrate the effectiveness of the proposed methods.

**Strengths:**

1. The proposed topology-aware aggregation strategies (Degree and Betweenness) are intuitive, simple to implement, taking in account the effect of topology on the learning effect.

2. The authors conduct experiments across five different datasets and multiple varied topologies and systematically study the impact of topology degree, modularity, and node count, providing a comprehensive characterization of the solution's performance.

**Weaknesses:**

1. The main concern is about the assumption of the data distribution and OOD data definition.I suspect the practicality of only one node having the OOD data. Besides, the experimental setup defines OOD data by a "backdoor" methodology (i.e., inserting triggers), as mentioned in Appendix B.2.2. This setup is very narrow and seems artificial. Also, this definition is different from the commonly used term OOD. The findings in this work may be hard to generalize to more natural setups.

2. The topology in this work is assumed to be static. However, in practice, the connection in IoT or edge computing networks is usually dynamic. When the topology changes, the definition of Betweenness may also vary.

3. This work relies on a synchronous communication model, which is a strong assumtion that does not reflect many real-world decentralized system.

**Questions:**

1. Given that the OOD data is defined as a backdoor, how can the method generalize to natural OOD scenarios, such as a node holding an entirely new data class?

2. If OOD data is not concentrated on a single node but is sparsely distributed across multiple random nodes, would the topology-aware strategy still be effective?

3. Since the method accelerates the propagation of all knowledge, how does it prevent being exploited to accelerate the spread of malicious backdoor or model poisoning attacks?

4. Did the study measure the impact of node churn (joining and leaving), which is critical for IoT or edge computing scenarios, on OOD knowledge propagation?

---

> ### Author Response · Authors · 2025-11-19
>
> Thank you for your review. We are glad you find our solution to be intuitive and simple to implement.
> - **W1/Q1/Q2 More general Non-IID/more non-IID sources**: First, we emphasize that our initial experimental design with a single controlled source of OOD information enabled us to exactly track how knowledge flows across devices. Having more than one source of OOD data makes it hard to distinguish the impact of topology vs. the impact of data on knowledge spread. Despite this, we run additional experiments to clarify the effect of more realistic non-IID data heterogeneity found in real-world settings by varying αₗ (distribution of label counts); the parameters governing data distribution are detailed in Appendix B.2.1. As (αₗ) goes to infinity, data distributions become more homogeneous; in this setting, there are multiple sources of OOD data (varies depending on αₗ). Please see Appendix H, Figure 21 for full experiment details and results. In summary, we find that at all levels of heterogeneity (multiple sources of OOD data) under more realistic conditions like label-skew, topology-aware methods outperform their topology-unaware counterparts.
> - **W2 Negligible overhead in Dynamic Topologies**: The cost of computing degree is O(1) and therefore negligible in both static and dynamic topologies. We detail the overhead of dynamically computing centrality metrics in Appendix C. Specifically in appendix C1. We profile the cost of computing betweenness and note that for graphs < 1000 nodes the cost is less than 1 second, and for 1024 nodes the cost is 2.35 seconds; **relative to the cost of training/communicating updates between nodes, this is negligible**. In static graphs, this is a **one time cost** incurred at the beginning of training which amortizes out over the course of training. The majority of time spent during training is for on-device training (detailed in Appendix D: Tables 4, 5). In the event that the topology is dynamically changing, **this cost would be incurred once** every time the topology changes. Given that Betweenness and Degree yield similar performance in real-world graphs (Figure 4,6), cheaper metrics like Degree should be used if the cost of Betweenness is prohibitive.
> - **Q3 Malicious actors**: The spread of corrupted or adversarial data is indeed a concern of **all** decentralized (and centralized) learning settings as we note in lines Appendix F in our limitations section. **This is not a limitation of our work, but rather an important consideration when designing any model training workflow.** Any robust training workflow needs to account for the possibility of malicious actors when collecting their training data. As with any machine learning setting, it is important to have safeguards to detect and remove unwanted training data. There are many approaches to detecting and spreading the spread of corrupted/malicious data. However, this study is the first of its kind: a large-scale empirical analysis of knowledge propagation in realistic real-world topologies. Effectively propagating knowledge in the first place remains an open problem (especially in the OOD setting). Accounting for corrupted/adversarial training data is important and outside scope of this investigation.
> - **W3 Adapting to asynchronous communication model**: A synchronous model is common. However, it is easy to adapt topology-aware aggregation techniques to the asynchronous settings; you would only need to wait for neighbors before each aggregation round. As this the first work to purpose topology-aware learning, we focus on first characterizing the performance of topology-aware methods in the synchronous settings and leave it to future work to explore the asynchronous setting.
> - **Q4 Knowledge Propagation in Dynamic Topologies**: We don’t study dynamic topologies here. But we expect the extension of topology-aware methods to be easy to implement in dynamic settings. We recommend using local centrality metrics in the case of fast evolving topologies which curbs the cost of computing global centrality metrics and adopting an asynchronous aggregation model in which a node simply waits for its neighbors. We leave this exciting direction for future work as investigating both the impact of topology on learning and aggregation is the focus of this work.

---

### Official Review · Reviewer_jBNa · 2025-11-01

**Soundness:** 2
**Presentation:** 2
**Contribution:** 2
**Rating:** 4
**Confidence:** 3

**Summary:**

This paper addresses the problem of propagating OOD knowledge across devices in decentralized learning. Unlike federated learning’s centralized aggregation, decentralized learning models only exchange updates with neighbors in a communication graph. This paper shows that existing topology-unaware aggregation methods struggle to disseminate OOD information: OOD accuracy lags IID accuracy by a large margin and is highly sensitive to both where OOD data originate and the network structure. To remedy this, this paper introduces simple topology-aware aggregation strategies that weight neighbor models by centrality metrics.

**Strengths:**

1. Clear identification of OOD propagation as a distinct and critical problem in decentralized learning.
2. Elegant, low-overhead methods that seamlessly integrate with existing gossip algorithms.
3. Comprehensive evaluation: five datasets, three topology families, varying OOD locations, and spectral-gap theory.

**Weaknesses:**

1. The paper assumes a single-device OOD “worst case” but does not analyze scenarios with multiple OOD sources, which may arise in practice and interact nonlinearly.
2. Only degree and betweenness are studied, yet other centrality metrics (e.g. eigenvector, closeness) might offer better trade-offs; appendix C claims negligible cost but no profiling of alternative metrics is shown.
3. Betweenness computation, even if amortized,scales superlinearly (O(nm + n² log n)), and while Table 3 reports <1 s for ≤1,000 nodes, real networks can be much larger; no discussion of dynamic topologies or incremental updates.
4. Hyperparameter τ is fixed at 0.1 for both metrics without ablation; it’s unclear how sensitive performance is to temperature choice.
5. The backdoor approach (10% red-square or token triggers) may not generalize to natural OOD shifts; more realistic distribution shifts (e.g. novel classes or domain changes) are not evaluated.
6. While memory/round messages stay the same, degree-based weighting implicitly assumes knowledge of global centrality, requiring initial full-graph exchange or central computation; the mechanism to share topology information securely is not detailed.
7. Beyond spectral-gap bounds, there is no formal convergence or generalization guarantee for non-convex objectives under topology-aware weighting, especially for heterogeneous data.
8. This work identifies and tackles OOD knowledge propagation in decentralized learning via two simple topology-aware aggregation schemes. The extensive empirical results validate their benefits, but broader applicability and scalability regarding multiple OOD sources, dynamic graphs, realistic shifts, and parameter sensitivity remain to be addressed.

**Questions:**

please see the weakness.

---

> ### Author Response · Authors · 2025-11-19
>
> Thank you for your review. We are glad you find our proposed method to be elegant and seamless to integrate with existing gossip algorithms.
> - **W1/W5 More general Non-IID/more non-IID sources**: First, we emphasize that our initial experimental design with a single controlled source of OOD information enabled us to exactly track how knowledge flows across devices. Having more than one source of OOD data makes it hard to distinguish the impact of topology vs. the impact of data on knowledge spread. Despite this, we run additional experiments to clarify the effect of more realistic non-IID data heterogeneity found in real-world settings by varying αₗ (distribution of label counts); the parameters governing data distribution are detailed in Appendix B.2.1. As (αₗ) goes to infinity, data distributions become more homogeneous; in this setting, there are multiple sources of OOD data (varies depending on αₗ). Please see Appendix H, Figure 21 for full experiment details and results. In summary, we find that at all levels of heterogeneity (multiple sources of OOD data) under more realistic conditions like label-skew, topology-aware methods outperform their topology-unaware counterparts.
> - **W2 Choice of centrality metrics**:  We discuss additional centrality metrics on lines 239-245. We choose degree and Betweenness as they capture local and global phenomena, respectively, and together expose a wide spectrum of knowledge propagation behavior. As this is the first exploratory study into topology-aware aggregation, we leave analysis of additional metrics to future work.  We instead prioritize a comprehensive study of both Degree/Betweenness. Through our experiments, Betweenness and Degree metrics of individual nodes do not substantially differ from one another. This is empirically validated in our work (see Figures 4, 6), where we observe very similar knowledge propagation performance between degree and Betweenness; in both cases topology-aware methods substantially outperform topology-unaware baselines. Therefore, we posit that our choice of centrality metrics for this initial work, under realistic real-world topology choices, is sufficient for arguing the necessity for topology-aware aggregation strategies.
> - **W3 Negligible overhead in Dynamic Topologies**: The cost of computing degree is O(1) and therefore negligible in both static and dynamic topologies. We detail the overhead of dynamically computing centrality metrics in Appendix C. Specifically in appendix C1. We profile the cost of computing betweenness and note that for graphs < 1000 nodes the cost is less than 1 second, and for 1024 nodes the cost is 2.35 seconds; **relative to the cost of training/communicating updates between nodes, this is negligible**. In static graphs, this is a **one time cost** incurred at the beginning of training which amortizes out over the course of training. The majority of time spent during training is for on-device training (detailed in Appendix D: Tables 4, 5). In the event that the topology is dynamically changing, **this cost would be incurred once** every time the topology changes. Given that Betweenness and Degree yield similar performance in real-world graphs (Figure 4,6), cheaper metrics like Degree should be used if the cost of Betweenness is prohibitive.
> - **W4 Tuning τ**: We add an extra analysis of tuning the temperature parameter in Appendix I. We notice that τ = 0.1 has a reasonable performance tradeoff between OOD AUC and IID AUC (see Fig 22).
> - **W7 Convergence Guarantees**: Because our decentralized learning problem is posed as a traditional gossip learning framework (but with topology-aware mixing coefficients), previously established convergence bounds for non-convex, non-IID optimization apply [1] (Appendix D).
> - **W6 Computing Centrality metrics:** The statement “degree-based weighting implicitly assumes knowledge of global centrality, requiring initial full-graph exchange or central computation” is **incorrect**; degree-based weighting only requires knowledge of local neighbors and can be computed in O(1) time without full-graph exchange. Further, mapping a networking topology is a common practice in networking (e.g., IoT networks, Ad-hoc networks). Therefore, from any given node, it is possible to map the entire topology using off-the-shelf network discovery protocols for both static [2] and dynamic [3] topologies.
>
> [1] A Unified Theory of Decentralized SGD with Changing Topology and Local Updates, ICML, 2020.
>
> [2] A Lightweight Network Discovery Algorithm for Resource-constrained IoT Devices, IEEE CNC.
>
> [3] Neighbor Discovery in Mobile Ad Hoc Networks Using an Abstract MAC Layer, IEEE Annual Allerton Conference.

---

> > ### Comment · Reviewer_jBNa · 2025-11-22
> > **Thanks for your rebuttal.**
> >
> > Thank you for your thoughtful rebuttal. My main concerns have been addressed. I will increase my rating score.

---

### Meta-Review · Area_Chair_X6wa · 2025-12-26

**Summary:**

This paper studies knowledge propagation in decentralized learning over arbitrary communication topologies, with a focus on how rare or out-of-distribution information spreads when models only aggregate with neighbors. The paper provides a broad empirical characterization showing that propagation can be strongly affected by both the topology and the placement of the rare data source. The paper proposes topology-aware aggregation weights based on degree or "betweenness", and shows consistent gains in the targeted propagation metric across multiple datasets and topology families.

I agree with the reviewers that this is a significant and timely issue. At the same time, because mixing rates and weight design in gossip and consensus are classic and well-studied, the paper would benefit from connecting its proposal more directly to that literature and from providing some theoretical treatment of the proposed weighting scheme, even if limited in scope. In the current version, the justification remains largely empirical, with only supporting intuition, and this is a central reason the work is criticized by the reviewers. Reviewer concerns about the representativeness of the trigger-based out-of-distribution setup and about static, synchronous assumptions further reduce confidence in the generality of the conclusions.

**Reviewer Concerns:**

The rebuttal addresses several concrete issues raised in the reviews. It adds experiments under more realistic heterogeneity (moving beyond a single rare-data source), includes sensitivity analysis for the temperature parameter, clarifies that degree-based weighting is locally computable, and discusses the overhead of recomputing centrality metrics. These additions likely resolve much of Reviewer jBNa’s practical concerns, and they partially address questions from Reviewer jFih about definitions, metric choice, and overhead.

The main issues driving the decision remain outstanding for the more skeptical reviewers. Reviewer Esqg’s concern about whether the conclusions extend beyond the trigger-based out-of-distribution setup is not fully resolved, and the paper still does not provide evidence on a natural shift (for example, new classes or domain changes). Reviewer ggNV continues to request a clearer theoretical treatment, and their follow-up highlights that the current appendix does not provide an explicit, self-contained bound or a clear reduction argument tailored to the proposed method. More broadly, the paper would be strengthened by a tighter positioning in the classical consensus and fastest-mixing literature, plus at least one theorem-level statement or proof sketch connecting the proposed coefficients to improved mixing or convergence behavior.

**Reviewer Scores:**

Reviewer jBNa explicitly stated after the rebuttal that their main concerns were addressed and that they would increase their rating.

Reviewer jFih raised concerns that the rebuttal partly addresses, so a modest increase seems plausible.

In contrast, the central concerns raised by reviewers Esqg and ggNV remain largely outstanding in the current state of the discussion, so I would not expect their scores to increase.

---

### Decision · Program_Chairs · 2026-01-26

Reject